# Decision Aggregation under Quantal Response

**Zhihuan Huang, Yichong Xia & Yuqing Kong**
Center on Frontiers of Computing Studies (CFCS)
School of Computer Science, Peking University
{zhihuan.huang, xiayc, yuqing.kong}@pku.edu.cn

## Abstract

The effectiveness of collective decision-making is often challenged by the bounded rationality and inherent stochasticity of individual agents. We investigate this by analyzing how to aggregate decisions from $n$ experts, each receiving a private signal about an unknown state. Assuming signals are conditionally independent and identically distributed, we depart from the fully rational paradigm and model expert behavior using quantal response—a stochastic choice model capturing bounded rationality. Within a minimax regret framework, we show that majority voting is the optimal robust aggregator when individual rationality falls below a certain threshold. Interestingly, such groups can outperform perfectly rational agents, as their decision randomness encodes weak but informative signals lost in deterministic behavior. We validate these findings using large language models (LLMs), which naturally exhibit quantal response via their temperature parameter. Aggregating moderately stochastic LLM outputs significantly improves accuracy on complex reasoning tasks, highlighting bounded rationality not as a limitation, but as a potential strength in collective intelligence.

## 1 Introduction

The market is very up and down. Your finger hovers over the BUY/SELL button. Seeking guidance, you consult two experts. Mia, renowned for her perfectly rational analysis, advises: "Sell." John, known for his sharp instincts but less predictable approach, urges: "Buy it. Trust me." Obviously, you should follow rational Mia. But what if you could consult multiple Mias or Johns?

Let's frame this formally. Suppose we are deciding whether to BUY ($X = 1$) or SELL ($X = 0$) a stock, where the true state $\omega$ is either rise ($\omega = 1$) or fall ($\omega = 0$). The utility function is defined as $u(X, \omega) = 1$ if $X = \omega$, and $u(X, \omega) = -1$ otherwise. We have two types of experts: Mia, who embodies perfect rationality, and John, who exhibits bounded rationality. Both receive private signals $S \in \{0, 1\}$ that provide noisy information about the future state. Their decisions depend on the expected utility of buying conditioning on private signals, denoted by $\mathbb{E}[u(1, \omega) \mid S]$, and follow a quantal response function (McKelvey and Palfrey, 1995):

$$\text{Buy Probability} = \Pr[\underbrace{X = 1}_{\text{buy}} \mid \underbrace{\mathbb{E}[u(1, \omega) \mid S] = v}_{\text{expected utility of buying is } v}] = \frac{e^{\lambda \cdot \mathbb{E}[u(1,\omega)|S]}}{\sum_{x \in \{0,1\}} e^{\lambda \cdot \mathbb{E}[u(x,\omega)]}}$$

$$= \frac{e^{\lambda v}}{e^{-\lambda v} + e^{\lambda v}} = \frac{1}{1 + e^{-2\lambda v}}$$

Let $\varphi_\lambda(v) = \frac{1}{1+e^{-2\lambda v}}$. $\varphi_\lambda(v)$ is the probability of buying given the advisor's expected payoff from buying ($X = 1$) is $v$, with a rationality parameter $\lambda$. In this scenario, "good" and "bad" signals correspond to expected payoffs of $v = -0.2$ (still bad, but less severe) and $v = -1$ (strongly avoid), respectively. Mia (Perfectly Rational, $\lambda \to \infty$) always follows strict logic—if

$v < 0$, she never buys; if $v > 0$, she always buys. John's (Bounded Rational, $\lambda = 2.5$) decisions are probabilistic—even when $v$ suggests "don't buy," he might still take a chance.

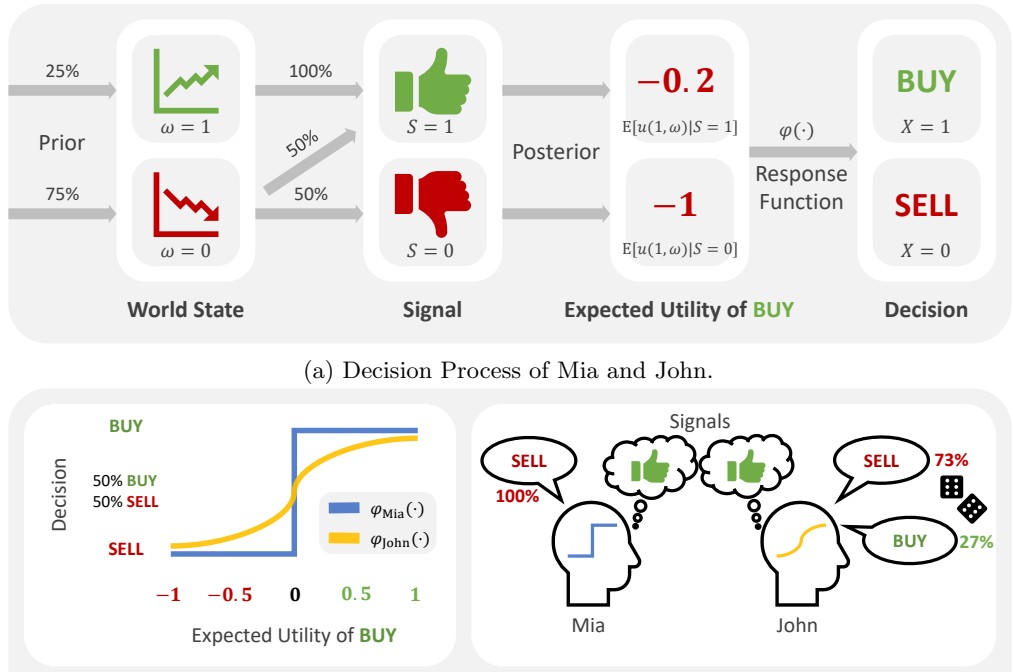

(a) Decision Process of Mia and John.

(b) Response Functions $\varphi_\lambda(v)$ for different $\lambda$.

Figure 1: Comparison of rational (Mia) vs. bounded-rational (John) decision-making through unified quantal response functions.

Mia, the perfectly rational expert, makes the "correct" choice every time—and earns an average utility of 0.5. John, who sometimes gambles against the odds, earns lower at 0.43.

Now, what if you ask a bunch of Mias and a bunch of Johns? Let's say each expert gets a private signal conditionally independent on the underlying state. Interestingly, after perfectly combining a team's judgments, a team of Mias stays at 0.5 (always right, but never better), while a team of Johns actually improves to 0.51—outperforming the perfect experts. Why? Because John's occasional "mistakes" add information for the group's collective wisdom to uncover deeper truths.

Our example shows how imperfect experts can outperform perfect ones - but there's a catch. To perfectly combine a team's judgments, you'd need to know their signal structures-joint distribution over signals and state. In reality, we almost never have this complete picture. This leads to our core question: How can we combine expert decisions effectively without knowledge of signal structures? Can imperfect experts still outperform perfect ones in this scenario?

This drives our research into robust aggregation methods that work without perfect information. We use a "worst-case scenario" framework. We assess different methods for combining opinions by evaluating their maximum "regret"—how much worse they could be compared to the best possible method if we did know everything (Arieli et al., 2018; 2023; Levy and Razin, 2022; De Oliveira et al., 2021). Surprisingly, within this framework, we find that simply using majority vote works best when experts aren't perfectly rational and exhibit sufficiently bounded rationality. Interestingly, majority vote isn't the best when all experts are perfectly rational. Furthermore, imperfect experts, when their opinions are simply tallied by majority vote, can actually do better than perfect experts whose opinions are combined using the most sophisticated techniques.

This advantage of bounded rationality aligns with observations in artificial intelligence: large language models (LLMs) exhibit similar quantal response behavior through their 'temperature' parameter. Lower temperatures produce deterministic, Mia-like outputs, while higher temperatures induce John-like stochasticity. Just as in human groups, lower-temperature outputs may yield individually accurate responses, but higher-temperature outputs leverage informational diversity to achieve superior collective decisions.

## 1.1 SUMMARY OF RESULTS

We study the aggregation of decisions from groups of boundedly rational experts modeled through quantal response functions. We consider experts receiving private signals that are conditionally independent and identically distributed (i.i.d.) given the true state. Our main contributions in Section 3 are as follows:

- **Optimal Robust Aggregator:** We prove majority voting is optimally robust for groups whose rationality falls below a threshold dependent on group size (Section A.2).
- **Bounded Rationality Advantage:** For a single expert ($n = 1$), rationality maximizes aggregation outcomes. However, with multiple experts ($n \geq 2$), bounded rationality can outperform perfect rationality (Section A.3). Generalizing beyond specific scenario, numerical experiments (Figure 5) show that bounded rational experts consistently achieve lower worst-case regret than fully rational experts.

These theoretical insights rely on a dimension reduction result: any quantal response report structure under conditional independence can equivalently be represented by a simplified three-signal structure, significantly reducing analytical complexity.

In Section 4, we empirically validate our theoretical predictions using large language models (LLMs) as experts. The 'temperature' parameter, which adjusts randomness, naturally corresponds to quantal response behavior.

- **Quantal Response in LLMs:** Empirical analysis confirms LLMs exhibit quantal response behavior, with logistic regression aligning closely with theoretical models (Figure 2).
- **Empirical Aggregation Advantage:** Using LLMs with varied temperatures ($t \in \{0, 0.5, 1\}$) and aggregating decisions via majority voting with groups of size $n = \{1, 3, 5\}$, we conduct two experiments: traditional box-ball tasks and multi-choice mathematical questions. Results confirm that bounded-rational experts with moderate temperature (e.g., $t = 0.5$) collectively outperform purely rational experts ($t = 0$) for $n \geq 3$, even in non-binary settings (Figure 3).

Overall, our findings reveal that following the majority decisions of moderately irrational experts is effective, highlighting benefits of bounded rationality. This insight impacts fields such as crowdsourcing, AI ensembles, economic decision-making, and product design, where incorporating diverse and less rational inputs can yield superior outcomes.

## 1.2 RELATED WORK

**Information Aggregation and Robustness.** Information aggregation—the process of combining multiple information sources into a collective decision—is fundamental to collective intelligence (Forsythe and Lundholm, 1990; Breiman, 1996; Dietterich, 2000). While traditional models often assume perfect rationality and complete information (Winkler, 1981; Arieli et al., 2018; 2023), real-world experts frequently deviate due to cognitive limitations and inherent stochasticity (Simon, 1990; Tversky et al., 1982). This reality necessitates robust aggregation methods capable of handling uncertain information structures. Recent literature has significantly expanded the frontiers of robust aggregation to tackle unknown correlations (Levy and Razin, 2022; De Oliveira et al., 2021), algorithmic feasibility (Guo et al., 2025), second-order information in two-expert scenarios (Pan et al., 2024), and adversarial expert inputs (Guo and Kong, 2025). Unlike prior studies that primarily focus on perfectly

rational or adversarial agents, we incorporate Quantal Response (QR) models (McKelvey and Palfrey, 1995; 1998) to capture bounded rationality through stochastic choice rules. By leveraging geometric dimension reduction, we prove that three signals suffice for optimal robust aggregation, explicitly demonstrating how bounded rationality can paradoxically benefit collective outcomes.

**Bounded Rationality and Strategic Voting.** Bounded rationality models have found extensive applications across mechanism design (Braverman et al., 2018), information design (Yu et al., 2024), and robust decision-making—where cognitive biases, such as base rate neglect, have been shown to yield surprising aggregation benefits (Kong et al., 2024; Gan et al., 2023; Feng et al., 2024). Specifically, QR models are widely used to analyze strategic behavior in probabilistic voting and collective choice (Goeree et al., 2002; 2016; Hoppe and Schmitz, 2013). For instance, Quantal Response Equilibrium (QRE) has been applied to examine voter strategies, laboratory voting games, and the "wisdom of strategic voting" (McKelvey and Patty, 2006; Casella et al., 2006; Goeree et al., 2005; Han et al., 2023). However, this rich literature typically centers on utility-based strategic manipulation where agents possess heterogeneous, outcome-dependent payoffs. In contrast, our research operates within a shared-objective framework where experts lack strategic incentives to misreport. We focus entirely on the mechanism of collective intelligence, aggregating sincere-but-bounded QR-experts rather than navigating strategic utility landscapes.

**Quantal Responses and Large Language Models.** The emergence of Large Language Models (LLMs) has spurred the application of behavioral economic models to AI agents (Jia et al., 2025; Kirshner et al., 2025). Concurrently, research on LLM temperature—which inherently controls output randomness—has investigated its impact on precision tasks (Renze and Guven, 2024), code generation (Zhu et al., 2024), and creativity (Peeperkorn et al., 2024). Existing works mostly study how a single LLM behaves or how LLMs interact competitively. Differently, we formally link the temperature parameter to QR theory within an aggregation context. We demonstrate that, much like human bounded rationality, optimally tuned LLM stochasticity can significantly improve the accuracy of aggregated group decisions.

## 2 MODEL SETUP

We consider a binary state $\Omega = \{0, 1\}$ with a known prior probability $\mu = \Pr[\omega = 1] \in [0, 1]$.

There is one decision maker (DM) and $n$ experts indexed by $i = 1, \ldots, n$. Each expert $i$ observes a private signal $S_i \in \mathcal{S}$, drawn from a set of possible signals $\mathcal{S}$, about the true state $\omega$. The joint distribution of the state and signals is called the *signal structure* $\theta \in \Delta(\Omega \times \mathcal{S}^n)$. The set of all possible signal structures is $\Theta$.

To capture scenarios such as aggregating multiple responses from a large language model—where each response is independently and identically generated from the same input—we focus on signal structures where signals are *conditionally independent and identically distributed (c.i.i.d.)* given the state $\omega$. Formally, this means:

$$\Pr[S_1 = s_1, \ldots, S_n = s_n \mid \omega = w] = \prod_{i=1}^{n} \Pr[S_i = s_i \mid \omega = w],$$

with the conditional distribution $\Pr[S_i = s \mid \omega = w]$ identical across all experts. The set of all such c.i.i.d. signal structures is denoted by $\Theta^{ciid}$. This setting is standard and describes situations where each expert independently observes information about the state. Additionally, since experts are anonymous from the DM's perspective, the DM treats each expert's signal distribution as identical.

Each expert $i$, after observing the private signal $S_i$, reports a binary decision $X_i \in \mathcal{X} = \{0, 1\}$ to the DM. The utility for both DM and experts depends on correctly identifying the state:

$$u(X, \omega) = \begin{cases} 1 & X = \omega \\ -1 & X \neq \omega \end{cases}.$$

This common payoff aligns their incentives towards correct decisions. We acknowledge that in strategic environments with heterogeneous utilities or pivotal incentives, sincere voting is not necessarily a Nash equilibrium (Austen-Smith and Banks, 1996; Han et al., 2023). However, our information-aggregation setting features a shared-objective structure where experts receive no outcome-dependent payoffs. Under this setting, experts have no strategic incentive to misreport, making sincere reporting a natural and widely used modeling assumption (Arieli et al., 2018; Palley and Satopää, 2023). This assumption is particularly appropriate when the experts are Large Language Models (LLMs), which do not possess stable, agent-specific utilities and do not benefit strategically from manipulating outcomes.

Experts choose their decisions according to a quantal response function, reflecting bounded rationality. We treat this as an "as-if" model rather than assuming experts perform precise Bayesian calculations followed by utility-maximization with noise. The randomness may arise from cognitive noise, imperfect internal signal processing, or intrinsic neural stochasticity (Knill and Pouget, 2004). Let $p = \Pr[\omega = 1 \mid S_i = s_i]$ be an expert's posterior belief after observing signal $s_i$. The expected utility of choosing $X_i \in \{0, 1\}$ is:

$$\mathbb{E}[u(X_i, \omega) \mid S_i] = (2X_i - 1)(2p - 1) \in [-1, 1].$$

Recall the quantal response function $\varphi_\lambda : [-1, 1] \to [0, 1]$, with rationality level $\lambda \geq 0$, is given by the logistic (softmax) form:

$$\varphi_\lambda(v) = \frac{1}{1 + e^{-2\lambda v}}, \quad v \in [-1, 1].$$

Note that $\varphi_\lambda$ is equivalent to the softmax layer in large language models, showing the similarity between quantal response in decision theory and softmax outputs in machine learning. Replacing $v = \mathbb{E}[u(X_i = 1, \omega) \mid S_i] = 2p - 1$, the expert's decision probability becomes:

$$\psi_\lambda(p) = \varphi_\lambda(2p - 1) = \frac{1}{1 + e^{2\lambda(1 - 2p)}}, \quad p \in [0, 1].$$

Note that $\varphi_\lambda$ and $\psi_\lambda$ are equivalent representations; for convenience, we use $\psi_\lambda$ throughout the remainder of this paper.

The function $\psi_\lambda$ increases with $p$. When $\lambda = 0$, decisions are random; as $\lambda \to \infty$, decisions become deterministic:

$$\psi_{\lambda \to \infty}(p) = \begin{cases} 1 & p > 0.5 \\ 0.5 & p = 0.5 \\ 0 & p < 0.5 \end{cases}.$$

Given a signal structure $\theta$ and response function $\psi_\lambda$, the *report structure* $\hat{\theta} \in \Delta(\Omega \times \mathcal{X}^n)$ describes the joint distribution of the state and experts' reports. The mapping $\mathrm{rep}(\theta, \psi_\lambda) = \hat{\theta}$ defines the set of c.i.i.d. report structures:

$$\hat{\Theta}_\lambda^{ciid} = \{\hat{\theta} : \exists \theta \in \Theta^{ciid}, \ \mathrm{rep}(\theta, \psi_\lambda) = \hat{\theta}\}.$$

### 2.1 Decision Aggregation

The DM aggregates experts' decisions anonymously based only on the number of experts $X = \sum_{i=1}^n X_i$ reporting 1. An *aggregator function* $f : \{0, \ldots, n\} \to [0, 1]$ represents the DM's mixed strategy, where $f(x)$ is the probability the DM guesses state $\omega = 1$ given $x$ reports of 1.

The utility for aggregator $f$ under report structure $\hat{\theta}$ is:

$$U(f, \hat{\theta}) = \sum_{x=0}^n \Pr_{\hat{\theta}}[X = x](2 \Pr_{\hat{\theta}}[\omega = 1 \mid X = x] - 1)(2f(x) - 1).$$

**Omniscient Aggregator.** An ideal benchmark, the omniscient aggregator maximizes expected utility knowing the true report structure $\hat{\theta}$:

$$
\text{opt}_{\hat{\theta}}(x) = \begin{cases} 1, & \Pr_{\hat{\theta}}[\omega = 1 \mid X = x] > 0.5 \\ 0.5, & \Pr_{\hat{\theta}}[\omega = 1 \mid X = x] = 0.5 \\ 0, & \Pr_{\hat{\theta}}[\omega = 1 \mid X = x] < 0.5 \end{cases}.
$$

**Optimal Robust Aggregator.** When facing uncertainty about $\hat{\theta}$, the DM aims to minimize worst-case regret, defined as the difference in utility between an aggregator $f$ and the omniscient aggregator:

$$
R(f, \hat{\theta}) = U(\text{opt}_{\hat{\theta}}, \hat{\theta}) - U(f, \hat{\theta}).
$$

The optimal robust aggregator solves the min-max problem:

$$
\text{opt}_{\hat{\Theta}} \in \arg\min_f \max_{\hat{\theta} \in \hat{\Theta}} R(f, \hat{\theta}),
$$

thus ensuring robustness under uncertainty.

## 3  THEORETICAL RESULTS

This section introduces theoretical results for decision aggregation under bounded rationality. We focus on a setting where experts receive signals that are independent and identically distributed conditioning on the world state (c.i.i.d.).

**Theorem 3.1** (Main Theorem). *Consider a group of experts with c.i.i.d. signal structures $\Theta^{ciid}$ and quantal responses with rationality parameter $\lambda$.*

1. ***Optimal Robust Aggregator.*** *When $\lambda \leq g(n)$, majority voting $f^{maj}$ is the optimal robust aggregator:*

$$
\max_{\hat{\theta} \in \hat{\Theta}_{\lambda}^{ciid}} R(f^{maj}, \hat{\theta}) = \min_f \max_{\hat{\theta} \in \hat{\Theta}_{\lambda}^{ciid}} R(f, \hat{\theta})
$$

   *using the decision rule:*

$$
f^{maj}(x) = \begin{cases} 0, & x < n/2 \\ 1/2, & x = n/2 \\ 1, & x > n/2 \end{cases}
$$

   *The threshold function $g(n)$ is defined as the supremum of the set of $\lambda$ values that satisfy a specific inequality for all $q_0$ and $q_1$ within a given range. Formally,*

$$
g(n) = \sup\left\{\lambda \,\middle|\, \forall q_0, q_1 \text{ such that } \psi_{\lambda}(0) \leq q_0 \leq q_1 \leq 0.5, \text{ the following inequality holds:}\right.
$$

$$
\left. \frac{(q_1(1-q_1))^{\lfloor \frac{n-1}{2} \rfloor}(1-2q_1)}{\psi_{\lambda}(1) - q_1} \leq \frac{(q_0(1-q_0))^{\lfloor \frac{n-1}{2} \rfloor}(1-2q_0)}{\psi_{\lambda}(1) - q_0} \times \frac{2\lambda + \ln(1-q_0) - \ln(q_0)}{2\lambda - \ln(1-q_0) + \ln(q_0)}\right\}
$$

   *For $n \leq 2$, majority voting remains optimal for any finite $\lambda$, that is, the threshold is infinity $g(n) = \infty$.*

2. ***Bounded Rationality Advantage.*** *While perfect rationality $(\lambda \to \infty)$ maximizes single-expert performance $(n = 1)$, the group exhibits a different behavior:*

   - *For all $n \geq 2$, there exist $\theta^* \in \Theta^{ciid}$ and finite $\lambda^*$ such that under $\theta^*$, the optimal utility with bounded rationality level $\lambda^*$ strictly exceeds that with perfect rationality:*

$$
\max_f U\left(f, rep\left(\theta^*, \psi_{\lambda^*}\right)\right) > \max_f U\left(f, rep\left(\theta^*, \psi_{\infty}\right)\right)
$$

- *For all $n > 2$, this advantage of bounded rationality holds for majority voting:*
$$U\left(f^{maj}, rep\left(\theta^*, \psi_{\lambda^*}\right)\right) > \max_f U\left(f, rep\left(\theta^*, \psi_\infty\right)\right)$$

In the above theorem, $\hat{\Theta}_\lambda^{ciid}$ denotes the set of joint distributions over reported decisions induced by the c.i.i.d. signal structures $\Theta^{ciid}$ through the quantal response function $\psi_\lambda$, i.e., $\hat{\Theta}_\lambda^{ciid} = rep(\Theta^{ciid}, \psi_\lambda)$. The terms $\psi_\lambda(0)$ and $\psi_\lambda(1)$ represent the probabilities of reporting $X_i = 1$ when the posterior $\Pr[\omega = 1 \mid S_i]$ is 0 or 1, respectively. The inequality that defines $g(n)$ illustrates that, in expectation, reporting 1 is optimal when exactly $\left\lfloor \frac{n-1}{2} \right\rfloor$ experts report 0, based on the combination of the two symmetric report structures parameterized by $q_0$ and $q_1$.

**Proof Sketch of Theorem 3.1**   The proof is structured in three main steps, which is shown in Section A.

1. **Dimension Reduction.** To solve the min-max optimization over the intractable infinite-dimensional space of signal structures, we reduce its dimensionality. Since an expert's report depends only on their posterior belief $s \in [0, 1]$, we encode each posterior as a point on a curve in $\mathbb{R}^3$. Valid report structures form the convex hull of this curve. We prove a key geometric lemma that no four points on this curve are coplanar. Combined with Carathéodory's theorem, this shows any report structure can be represented by a signal structure with at most three posteriors $\{0, p, 1\}$, simplifying the problem to a manageable three-parameter space.

2. **Optimality of Majority Voting.** Bounded rationality regularizes the space of report structures, smoothing out extreme structures that favor complex, non-monotonic aggregators. We prove that when $\lambda \leq g(n)$, simple majority voting becomes the optimal robust aggregator. To establish this, we prove **pairwise optimality**: for any symmetric pair of structures, majority voting minimizes their combined regret. We then show this implies global minimax optimality. The threshold $g(n)$ is the largest $\lambda$ for which pairwise optimality holds.

3. **Bounded Rationality Advantage.** Finally, we explain the counterintuitive result that bounded rationality can outperform perfect rationality in a group setting ($n \geq 2$). While perfect rationality is best for a single expert, the randomness from bounded rationality introduces valuable diversity in a group. It allows weak but useful signals, which might be lost when fully rational experts all report the same decision, to influence the collective outcome. The proof is by construction. We design a signal structure $\theta^*$ where perfectly rational experts unanimously report 0, yielding a utility of 0.5. We then show that with a finite rationality level $\lambda^*$, their reports become stochastic. This variation, while noisy, is informative, allowing an aggregator to achieve a utility strictly greater than 0.5. For $n > 2$, we show that simple majority voting is sufficient to gain this advantage.

**Numerical Study.**   To complement our theoretical analysis, we conducted numerical simulations to illustrate the key results. As shown in Section A.2 (Figure 5), we computed the worst-case regret for both majority voting and the optimal robust aggregator. The results confirm that for rationality levels $\lambda \leq g(n)$, the regrets are identical, verifying that majority voting ($f^{maj}$) is indeed the optimal robust aggregator. The plots also reveal that this threshold is a sufficient but not strictly necessary condition, as majority voting can remain optimal for some $\lambda > g(n)$. However, for larger groups (e.g., $n = 5$) and high rationality, a more complex aggregator outperforms the majority rule. The U-shaped curve of the worst-case regret illustrates that choosing an appropriate, bounded level of rationality can be more effective than pursuing perfect rationality. This highlights that a moderate degree of "noise" or uncertainty can be beneficial for group decision-making.

## 4   Empirical Results

In this section, we describe two sets of studies designed to address the following two key research questions:

1. Do the responses of large language model (LLM) conform to *quantal response* theory?

2. How do the number of *experts* (LLM instances) and their *rationality* (temperature) affect the aggregation results, and how should these parameters be set to optimize performance?

**Connection between LLMs and Quantal Response.**   Before detailing the experimental setup, we formalize the connection between the quantal response (QR) model and the core output mechanism of LLMs. The QR model is structurally identical to the LLM's softmax function with temperature. Let $u_j = \mathbb{E}[u(X = j, \omega)|S_i]$ denote the expected utility of choosing action $j$. The QR model dictates that the probability of choosing action $j$ is given by:

$$P(X = j) = \frac{e^{\lambda u_j}}{\sum_k e^{\lambda u_k}}$$

where $\lambda$ is the rationality parameter. Similarly, the probability of an LLM outputting choice $j$ is determined by its internal logits $z_j$ and the sampling temperature $t$:

$$P(\text{choice } j) = \frac{e^{z_j/t}}{\sum_k e^{z_k/t}}$$

These formulations are structurally identical. Assuming the LLM's internal logits are proportional to the expected utility under a given task, the rationality parameter $\lambda$ is mathematically analogous to the inverse temperature $1/t$. A lower $\lambda$ (increased stochasticity) directly corresponds to a higher temperature $t$ (more randomness). This theoretical link makes LLMs ideal subjects for evaluating bounded-rational agents, an approach increasingly supported by recent literature modeling LLM decision-making through behavioral game theory and QR frameworks (Jia et al., 2025; Kirshner et al., 2025).

These questions and theoretical connections guide the study setup. We simulate individual expert decisions by querying an LLM and using the single response as the expert's choice. To simulate the conditionally independent and identically distributed (c.i.i.d.) nature of expert decisions, we generate multiple responses to the same query independently. To balance performance and cost, we selected *gpt-4o-mini* as the model and used its API for querying.

## 4.1 Bayesian Decision-Making Study

The first study examines whether LLM responses follow quantal response theory in a Bayesian decision framework.

**Study Design**   We use the standard belief-updating task Phillips and Edwards (1966); Grether (1980) as the Bayesian Decision-Making Study. Specifically, large language models (LLMs) are tasked with inferring which of two boxes was selected based on the observed color of a ball drawn from the chosen box. The left box ($\omega = 1$) and the right box ($\omega = 0$) contain red and blue balls in different proportions. The prior probability of selecting the left box is $\mu = \Pr[\omega = 1]$, and the task is to infer the selected box based on the observed signal, $S_i$. The model is queried under three temperature settings (0, 0.5, and 1) to assess the impact of model rationality on decision-making. By discretization, we get 400 scenarios. Each scenario is repeated 20 times to capture stochastic variability, and the decision proportion is calculated as the empirical probability of choosing the left box across all repetitions. The details are provided in Section B.

**Examining the Quantal Response**   Logistic regression analysis was conducted to test whether the decision-making behavior of the LLM adheres to the quantal response model. The dependent variable is the model's decision proportion, while the independent variable is the posterior belief derived from the Bayesian framework. To ensure symmetry in the logistic regression, a symmetric processing approach was applied: for each observation (posterior, proportion), a corresponding symmetric observation (1-posterior, 1-proportion) was added. This transformation effectively doubled the dataset, expanding from 400 observations to 800, with each temperature condition. The regression coefficient $\lambda$ corresponds to the rationality

level in the quantal response, with higher values suggesting more deterministic responses and lower values reflecting increased randomness.

The results of the logistic regression, visualized in Figure 2, reveal a clear trend. As the temperature increases, the coefficient $\lambda$ decreases, signifying a transition from more deterministic to more probabilistic decision-making. Specifically, $\lambda$ ranges from a value close to infinity ($\lambda \to \infty$) at a temperature of 0.0, to 13.25 at 0.5, and further to 8.93 at 1.0. These results are statistically significant, as evidenced by z-values far exceeding the critical threshold of $\pm 1.96$ (for a 95% confidence level) and p-values effectively equal to 0 for all settings. These findings indicate that LLMs exhibit probabilistic decision-making consistent with the quantal response theory. A detailed summary of the regression results is provided in the appendix (see Table 1).

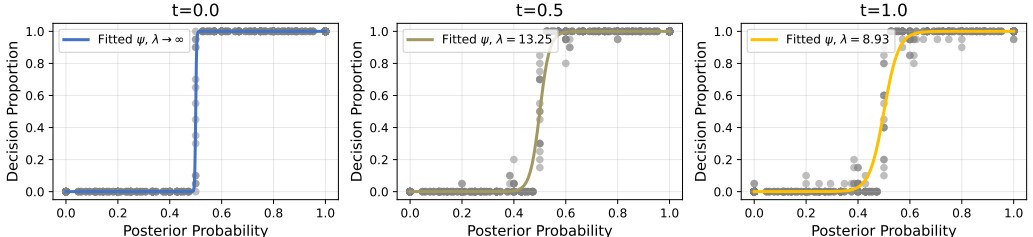

Figure 2: **Logistic Regression Results Across Temperatures.** The plots illustrate the relationship between posterior probability (x-axis) and decision proportion (y-axis) at three temperature settings ($t = 0.0, 0.5, 1.0$). Each gray dot represents a specific scenario, showing the decision proportion of *GPT-4o-mini* for that scenario. The fitted quantal response curve (orange line) represents the model's predicted behavior, with the fitted coefficient $\lambda$ quantifying the rationality of decision-making. Specifically, $\lambda \to \infty$ at $t = 0.0$, $\lambda = 13.25$ at $t = 0.5$, and $\lambda = 8.93$ at $t = 1.0$. The results demonstrate increased randomness as temperature rises, aligning with the quantal response model's predictions.

## 4.2 Multiple-Choice Question Answering Study

The second study investigates the behavior of large language models (LLMs) when tasked with multiple-choice question answering, using the MathQA dataset (Amini et al., 2019). This study focuses on understanding the efficacy of plurality vote aggregation for problems involving multiple options and evaluates performance under varying levels of model rationality and the number of experts.

**Study Design** We evaluate aggregation performance by querying the model with 500 randomly selected questions from the MathQA dataset, under three temperature settings ($t = 0.0, 0.5, 1.0$). For each question, we generate 20 responses. Because the questions offer more than two options, we use plurality voting ($f^{plu}$), a generalization of majority voting. This involves selecting the option with the highest frequency among the sampled responses. In the event of a tie, the aggregator randomly chooses among the tied options. Bootstrapping is used to assess aggregation robustness, repeated 1000 times for each temperature and expert group size ($n = 1, 3, 5$). Accuracy is the proportion of correct aggregated answers. The details are provided in Section B.

**Evaluate Performance of Aggregators** The performance of aggregation methods is evaluated under different temperature settings ($t = 0.0, 0.5, 1.0$) and varying numbers of experts ($n = 1, 3, 5$). Specifically, we examine majority voting ($f^{maj}$) in the Bayesian Decision-Making Study and plurality voting ($f^{plu}$) in the Multiple-Choice Question Answering Study using the MathQA dataset. We use only odd values for $n$ to avoid ties.

Across both studies (Figure 3), when $n = 1$, the expected utility is generally lower due to lack of decision diversity. Deterministic expert behavior ($t = 0.0$) yields slightly higher utility compared to stochastic settings ($t \geq 0.5$), indicating that with only one expert,

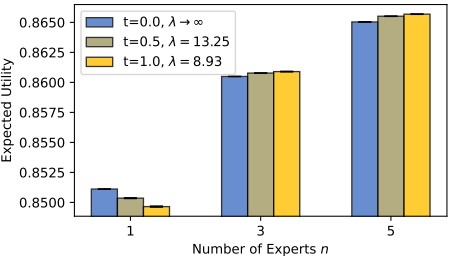 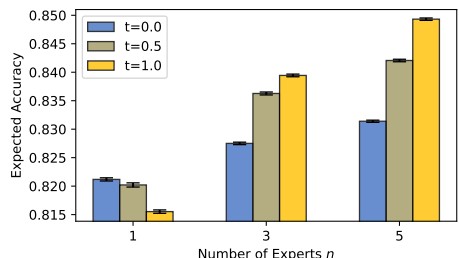

(a) Bayesian Decision-Making Study        (b) Multiple-Choice QA Study

Figure 3: **Performance of Majority/Plurality Voting Aggregation.** Panel (a) presents the expected utility of the majority-vote aggregator ($f^{maj}$) in the Bayesian decision-making task, where different temperature settings ($t = 0.0, 0.5, 1.0$) and numbers of experts ($n = 1, 3, 5$) are evaluated. Panel (b) shows the accuracy of the plurality-vote aggregator ($f^{plu}$) when applied to the multiple-choice question answering task using the MathQA dataset. In both panels, error bars represent the standard error of the mean (SEM), which quantifies the uncertainty in the estimated mean. Both studies show the same pattern: increasing the number of experts ($n$) improves aggregation performance. When $n = 1$, higher temperature ($t$) decreases performance due to increased randomness. When $n \geq 3$, higher temperature improves performance as diversity enhances aggregation effectiveness. These results highlight that while randomness degrades individual decisions, it benefits collective decision-making when properly aggregated, which aligns with our theoretical findings.

randomness negatively impacts decision reliability. However, when $n = 3, 5$, the utility of majority (plurality) voting interestingly increases with higher temperatures. This outcome aligns with our theoretical predictions, suggesting that randomness can enhance aggregation performance by providing additional informational diversity. Additionally, we evaluate a simple random follow rule, which performs equivalently to majority voting with $n = 1$, and find its performance consistently inferior or equal to majority voting with larger groups.

These findings align closely with theoretical expectations and ensemble learning literature, emphasizing the value of aggregating multiple stochastic responses from LLMs to improve decision accuracy.

## 5 Discussion and Future Work

This paper shows that bounded rationality plays a crucial role in shaping decision aggregation. Majority voting reaches minimax optimality when group rationality $\lambda$ is below a critical threshold ($g(n)$). In particular, experts with moderate levels of stochastic behavior ($\lambda < \infty$) sometimes outperform perfectly rational experts. This phenomenon can be explained theoretically in a robust aggregation framework and is observed empirically in large language models (LLMs), where temperature-controlled randomness improves the accuracy of aggregated decision. These results suggest that introducing imperfections may enhance collective intelligence.

While these findings offer practical implications for domains such as investment (where "noisy" retail investors are common) and AI ensembles (where temperature diversity can be optimized), several assumptions require relaxing for real-world applications. Our current framework assumes conditionally independent and identically distributed (i.i.d.) experts, neglecting potential correlations and adversarial conditions. Furthermore, the homogeneous rationality model does not account for groups with diverse rationality. Future research should address signal dependencies through graph-based aggregation methods and explore the impact of heterogeneous rationality on decision-making, including real-world testing in critical areas like medical diagnostics.

This work is supported by National Natural Science Foundation of China award number 62372007.

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

# A  Proofs for Section 3

## A.1  Dimension Reduction

To prove Section 3, the primary challenge is to reduce the complexity of the signal space. Since, by definition of the response function, the report depends only on the posterior given $\lambda$, we can slightly reduce complexity by grouping signals that yield identical posteriors. We use $\mathcal{S}_\theta$ to denote the set of signals that can be received by each expert under $\theta$. The above observation is formalized as follows:

**Observation A.1** (Signal Grouping). *Let* $\mathcal{S}_\theta = \{s^{(1)}, s^{(2)}, \ldots, s^{(m)}\}$ *be the signal set of the signal structure* $\theta \in \Theta^{ciid}$. *For any two signals* $s^{(j)}, s^{(k)} \in \mathcal{S}_\theta$ *with identical posteriors, i.e.,*

$$\Pr[\omega = 1 \mid S_i = s^{(j)}] = \Pr[\omega = 1 \mid S_i = s^{(k)}],$$

*we define a new signal* $s^{(j \vee k)}$ *representing the event* $S_i = s^{(j)}$ *or* $S_i = s^{(k)}$. *That is,*

$$S_i = s^{(j \vee k)} \iff S_i = s^{(j)} \ or \ S_i = s^{(k)}.$$

*By grouping such signals, we construct a new signal structure* $\theta'$ *with a smaller signal set that induces the same report structure as* $\theta$:

$$\forall \lambda > 0, \quad rep(\theta, \psi_\lambda) = rep(\theta', \psi_\lambda).$$

After grouping signals with identical posteriors, we establish a one-to-one correspondence between signals and posteriors. This allows us to represent each signal by its corresponding posterior value. For instance, a signal $s$ where $\Pr[\omega = 1 \mid S_i = s] = 0.8$ can be represented simply as 0.8.

Now, let's define a 3-signal space signal structure, denoted $\Theta^{3ciid}$. This structure comprises signal structures where the signal set contains at most three distinct posteriors:

$$\Theta^{3ciid} = \left\{ \theta \in \Theta^{ciid} \mid \exists p \in (0, 1) \text{ such that } \mathcal{S}_\theta \subseteq \{0, p, 1\} \right\}.$$

As defined, any $\theta \in \Theta^{3ciid}$ has a signal set $\mathcal{S}_\theta$ containing at most three elements, each representing a posterior. At most one of these posteriors, $p$, can lie strictly between 0 and 1; the others must be 0 and 1. We can parameterize each signal structure $\theta \in \Theta^{3ciid}$ with three values: $p$, $\gamma_0$, and $\gamma_1$. These parameters define the joint distribution:

| | | $S_i = 0$ | $S_i = p$ | $S_i = 1$ |
|---|---|---|---|---|
| $\theta$ | $\omega = 0$ | $\gamma_0$ | $\gamma_1(1-p)$ | $0$ |
| | $\omega = 1$ | $0$ | $\gamma_1 p$ | $1 - \gamma_0 - \gamma_1$ |

This 3-signal space structure represents a significant simplification compared to the general signal space. The following lemma will demonstrate the sufficiency of considering only this simplified structure.

**Lemma A.2** (Dimension Reduction). *For all quantal response rationality levels* $\lambda > 0$, *the report structure induced by the 3-signal space structure is equivalent to that induced by the general signal space. Formally:*

$$rep(\Theta^{3ciid}, \psi_\lambda) = \hat{\Theta}_\lambda^{ciid}$$

We now formalize the dimension reduction technique that allows us to compress the signal space.

**Proof Sketch of Lemma A.2.**

The proof consists of three main steps:

1. **3-Dimensional Encoding of Signal and Report Structures.**
   We begin by encoding signal structures and report structures into a 3-dimensional space. Under the response function $\psi_\lambda$, each signal $s \in [0, 1]$ corresponds to a point

$$\boldsymbol{v}(s) = (s, \psi_\lambda(s), s\psi_\lambda(s))$$
$$= (\Pr[\omega = 1 \mid S_i = s], \Pr[X_i = 1 \mid S_i = s], \Pr[X_i = 1, \omega = 1 \mid S_i = s])$$

in three-dimensional space. Here, $\psi_\lambda(s) = \Pr[X_i = 1 \mid S_i = s]$ represents the probability of taking action 1 given posterior $s$, while $s\psi_\lambda(s) = \Pr[X_i = 1, \omega = 1 \mid S_i = s]$ represents the joint probability of taking action 1 and the action matching the true state of the world. As $s$ varies over the interval $[0, 1]$, this mapping traces out a curve, which we denote by $\mathcal{C}$, in the three-dimensional space.

Each report structure also corresponds to a point in three-dimensional space:

$$\boldsymbol{v}(\hat{\theta}) = (\Pr[\omega = 1], \Pr[X_i = 1], \Pr[X_i = 1, \omega = 1]).$$

Critically, the point corresponding to a report structure $\hat{\theta} = \text{rep}(\theta, \psi_\lambda)$ is a convex combination of the points corresponding to the signals in the signal set $\mathcal{S}_\theta$ of $\theta$. Consequently, all possible report structures correspond to points within the convex hull of the curve $\mathcal{C}$.

Thus, to establish the equivalence of report structures induced by the 3-signal space and the general signal space, it suffices to show that their respective convex hulls are equivalent. Specifically, we must demonstrate that any point $\boldsymbol{v}(\hat{\theta})$ within the convex hull of $\mathcal{C}$ can be expressed as a convex combination of at most three points $\boldsymbol{v}(0), \boldsymbol{v}(p), \boldsymbol{v}(1)$ on $\mathcal{C}$, corresponding to signals of 0, $p$, and 1, for some $p \in [0, 1]$.

2. **No Four Points Coplanar.**
   To prove this, we need the following key result: no four points on the curve $\mathcal{C}$ are coplanar. We establish this by considering a 4x4 matrix where each row takes the form $[1, s, \psi_\lambda(s), s\psi_\lambda(s)]$ for four values of $s : a < b < c < d$. Assuming $\psi_\lambda(p)$ is the quantal response function, we demonstrate that the determinant of this matrix maintains a consistent sign (either always positive or always negative) regardless of the values of $a, b, c, d$, which implies the result.

3. **Representing Any Report Structure $\boldsymbol{v}(\hat{\theta})$ with $\boldsymbol{v}(0), \boldsymbol{v}(p), \boldsymbol{v}(1)$.**
   Using the "no four points coplanar" result, and by repeatedly applying it along with the intermediate value theorem (Rudin et al., 1964), we first establish a preliminary result: any point on the line segment connecting two points $\boldsymbol{v}(j), \boldsymbol{v}(k)$ on $\mathcal{C}$ can be expressed as a convex combination of at most three points $\boldsymbol{v}(0), \boldsymbol{v}(p), \boldsymbol{v}(1)$ on $\mathcal{C}$, corresponding to signals 0, $p$, and 1 for some $p \in [0, 1]$. We then extend this result to encompass any point in the convex hull of $\mathcal{C}$ through an iterative reduction argument.

*Proof of Lemma A.2.*

**3-Dimensional Encoding of Signal and Report Structures.**

We need to first analyze the relationship between the report structure and the signal structure.

To analyze the report structure $\hat{\theta}$ of a specific expert, we observe that three quantities can uniquely determine a report structure of expert $i$: $\Pr_{\hat{\theta}}[\omega = 1]$, $\Pr_{\hat{\theta}}[X_i = 1]$, and $\Pr_{\hat{\theta}}[X_i = 1, \omega = 1]$. These three quantities can be considered as a point in $\mathbb{R}^3$. We define $\boldsymbol{v}(\hat{\theta})$ as the point in $\mathbb{R}^3$ corresponding to $\hat{\theta}$. Formally,

$$\boldsymbol{v}(\hat{\theta}) = (\Pr_{\hat{\theta}}[\omega = 1], \Pr_{\hat{\theta}}[X_i = 1], \Pr_{\hat{\theta}}[X_i = 1, \omega = 1])$$

Using this perspective, we can establish the connection between the signal structure and the report structure. Consider a signal $s \in [0, 1]$ with posterior $\Pr[\omega = 1 \mid S_i = s] = s$. We represent this signal as a point in $\mathbb{R}^3$ using the following coordinates:

$$\begin{aligned}
\boldsymbol{v}(s) &= (s, \psi(s), s\psi(s)) \\
&= (\Pr[\omega = 1 \mid S_i = s], \psi(\Pr[\omega = 1 \mid S_i = s]), \Pr[\omega = 1 \mid S_i = s]\psi(\Pr[\omega = 1 \mid S_i = s])).
\end{aligned}$$

This point represents the report structure when only this specific signal $s$ is considered. Here, $\Pr[\omega = 1 \mid S_i = s]$ is the posterior probability that the true state is $\omega = 1$ given the signal $s$, $\psi(\Pr[\omega = 1 \mid S_i = s])$ is the probability that the expert reports 1 given this posterior, and $\Pr[\omega = 1 \mid S_i = s]\psi(\Pr[\omega = 1 \mid S_i = s])$ is the joint probability that $\omega = 1$ and the expert reports 1.

Given the signal $S_i = s$, the state $\omega$ and the decision $X_i$ are conditionally independent. This conditional independence allows us to express the overall report structure as a linear combination of the points corresponding to different signals.

**Lemma A.3.** *For corresponding signal structure $\theta$ and report structure $\hat{\theta}$, the point of report structure $\hat{\theta}$ for expert $i$ can be represented as a convex combination of the points corresponding to the individual signals $\boldsymbol{S}$ in the signal structure $\theta$. That is,*

$$\boldsymbol{v}(\hat{\theta}) = \sum_{s \in \mathcal{S}_\theta} \Pr_\theta[S_i = s]\boldsymbol{v}(s)$$

*Proof of Lemma A.3.* Let $\theta$ be the corresponding signal structure of $\hat{\theta}$, and $\mathcal{S}_\theta$ be the set of all possible signals $s$ that expert $i$ can receive. The probability of receiving signal $s$ is denoted by $\Pr_\theta[S_i = s]$. The three quantities defining the overall report structure can be expressed as expectations over these points:

- $\Pr_{\hat{\theta}}[\omega = 1]$ is the expected posterior probability:

$$\Pr_{\hat{\theta}}[\omega = 1] = \sum_{s \in \mathcal{S}_\theta} \Pr_\theta[S_i = s]\Pr_\theta[\omega = 1 \mid S_i = s].$$

- $\Pr_{\hat{\theta}}[X_i = 1]$ is the expected reporting probability:

$$\Pr_{\hat{\theta}}[X_i = 1] = \sum_{s \in \mathcal{S}_\theta} \Pr_\theta[X_i = 1, S_i = s]$$

$$= \sum_{s \in \mathcal{S}_\theta} \Pr_\theta[S_i = s]\psi(\Pr_\theta[\omega = 1 \mid S_i = s]).$$

- $\Pr_{\hat{\theta}}[X_i = 1, \omega = 1]$ is the expected joint probability:

$$\Pr_{\hat{\theta}}[X_i = 1, \omega = 1] = \sum_{s \in \mathcal{S}_\theta} \Pr_\theta[X_i = 1, S_i = s, \omega = 1]$$

$$= \sum_{s \in \mathcal{S}_\theta} \Pr_\theta[S_i = s]\Pr_\theta[X_i = 1, \omega = 1 \mid S_i = s]$$

$$= \sum_{s \in \mathcal{S}_\theta} \Pr_\theta[S_i = s]\Pr_\theta[\omega = 1 \mid S_i = s]\Pr_\theta[X_i = 1 \mid S_i = s]$$

$$\text{(Conditional Independence)}$$

$$= \sum_{s \in \mathcal{S}_\theta} \Pr_\theta[S_i = s]\Pr_\theta[\omega = 1 \mid S_i = s]\psi(\Pr_\theta[\omega = 1 \mid S_i = s]).$$

Each of these expectations is a weighted sum of the corresponding coordinates of the points $\boldsymbol{v}(s)$ in $\mathbb{R}^3$, with weights given by the probabilities $\Pr_\theta[S_i = s]$. Thus, the point of report structure $\boldsymbol{v}(\hat{\theta})$ is a convex combination of the points $\boldsymbol{v}(s)$.

$$\boldsymbol{v}(\hat{\theta}) = \sum_{s \in \mathcal{S}_\theta} \Pr_\theta[S_i = s]\boldsymbol{v}(s)$$

$\square$

Define curve $\mathcal{C}$ be the set of all points of signals in $\mathbb{R}^3$, that is, $\mathcal{C} = \{(s, \psi(s), s\psi(s)) \mid s \in [0, 1]\}$. Let $conv(\mathcal{C})$ be the convex hull of curve $\mathcal{C}$. Since the point of report structure can be represented as a convex combination of the points in $\mathcal{C}$, $conv(\mathcal{C})$ is the set of all valid report structures.

According to Carathéodory (1911)'s theorem, in $\mathbb{R}^3$, any point inside the convex hull of a set can be represented as a convex combination of at most 4 points from the set. Thus, we only

need 4 signals to represent all report structures. However, in the next subsection, we will further prove that when $\psi(s)$ is a quantal response function, the signal structures in $\Theta^{3ciid}$ can express all report structures.

**No Four Points Coplanar.**

Then we prove that, given $\lambda > 0$, there are no four points coplanar in the curve $\mathcal{C} = \{(s, \psi_\lambda(s), s\psi_\lambda(s)) \mid s \in [0,1]\}$. Formally,

**Lemma A.4.** *Given $\lambda > 0$, any four different points on*

$$\boldsymbol{v}(s) = (s, \psi_\lambda(s), s\psi_\lambda(s)), (s \in [0,1])$$

*are not coplanar.*

*Proof of Lemma A.4.* Let $M(a,b,c,d) = \begin{pmatrix} 1 & a & \psi_\lambda(a) & a\psi_\lambda(a) \\ 1 & b & \psi_\lambda(b) & b\psi_\lambda(b) \\ 1 & c & \psi_\lambda(c) & c\psi_\lambda(c) \\ 1 & d & \psi_\lambda(d) & d\psi_\lambda(d) \end{pmatrix}$.

We need to prove $\forall a,b,c,d, 0 \le a < b < c < d \le 1$,

$$\det(M(a,b,c,d)) > 0.$$

Let $\beta = 4\lambda$, $u = d - c$, $v = c - b$, $w = b - a$,

$$h(x) = \begin{cases} \beta, & x = 0 \\ \frac{e^{\beta x} - 1}{xe^{\frac{\beta x}{2}}}, & x > 0 \end{cases}, \quad g(x) = \begin{cases} \ln \beta, & x = 0 \\ \ln(e^{\beta x} - 1) - \ln x, & x > 0 \end{cases},$$

Then we have

$$\det(M(a,b,c,d)) > 0$$
$$\iff e^{2(2a+2b+2c+2d-2)\lambda} \prod (1 + e^{2\lambda(1-2a)}) \det(M(a,b,c,d)) > 0$$
$$\iff (d-c)(b-a)(e^{d\beta} - e^{a\beta})(e^{c\beta} - e^{b\beta}) > (d-a)(c-b)(e^{d\beta} - e^{c\beta})(e^{b\beta} - e^{a\beta})$$
$$\iff uw(e^{(u+v+w)\beta} - 1)(e^{v\beta} - 1) > v(u+v+w)(e^{w\beta} - 1)(e^{u\beta} - 1)e^{v\beta}$$
$$\iff h(v)h(u+v+w) > h(u)h(w)$$

When $x > 0$, we define $t = \beta x > 0$.

Then

$$f'(x) > 0$$
$$\iff \frac{\beta}{2}xe^{\frac{3\beta x}{2}} + \frac{\beta}{2}xe^{\frac{\beta x}{2}} - e^{\frac{3\beta x}{2}} + e^{\frac{\beta x}{2}} > 0$$
$$\iff \beta xe^{\beta x} + \beta x - 2e^{\beta x} + 2 > 0$$
$$\iff te^t + t - 2e^t + 2 > 0$$
$$\iff \sum_{k=2}^{+\infty} \frac{t^k(k-2)}{k!} > 0$$

Then we know $h(u+v+w) > h(u+w)$ and $h(v) > h(0) = \beta$. Since $\lim_{x \to 0^+} h(x) = g(0)$, $h(x)$ is continuous when $x \ge 0$. Because $\forall x \ge 0, h(x) > 0$,

$$h(v)h(u+v+w) > h(u)h(w)$$
$$\impliedby h(u+w)h(0) \ge h(u)h(w)$$
$$\iff \ln(h(u+w)) + \ln(h(0)) \ge \ln(h(u)) + \ln(h(w))$$
$$\iff g(u+w) + g(0) \ge g(u) + g(w)$$

Since $\lim_{x \to 0^+} g(x) = g(0)$, $g(x)$ is continuous when $x \ge 0$. When $x > 0$, we define $m = \beta x > 0$.

Then

$$g''(x) > 0$$

$$\iff \frac{1}{x^2} - \frac{\beta^2 e^{\beta x}}{(e^{\beta x} - 1)^2} > 0$$

$$\iff (e^m - 1)^2 > m^2 e^m$$

$$\iff e^m + e^{-m} - 2 > m^2$$

$$\iff \sum_{k=2}^{+\infty} \frac{2m^{2k}}{(2k)!} > 0$$

Then we know $g(x)$ is convex on $[0, +\infty)$. So $g(u + w) + g(0) \geq g(u) + g(w)$, which means that the theorem is proved. $\square$

**Representing Any Report Structure $v(\hat{\theta})$ with $v(0), v(p), v(1)$.**

In the following Lemmas A.5 and A.6, we will first show that for any convex combination of two points on the curve, the resulting point can be expressed as a convex combination of the points $v(0), v(p), v(1)$.

**Lemma A.5.** *When $\psi_\lambda(p)$ is a continuous function such that no four distinct points on the curve $\mathcal{C} = \{(s, \psi(s), s\psi(s)) \mid s \in [0, 1]\}$ are coplanar. Then, $\forall 0 < p_1 < p_2 < 1, 0 \leq q \leq 1$, s.t.*

$$q v(p_1) + (1 - q) v(p_2), v(p), v(0), v(1) \text{ are coplanar.} \tag{1}$$

*There is only one $p \in [0, 1]$ satisfies eq. (1), and $p \in [p_1, p_2]$*

*Proof of Lemma A.5.* Let $M(a, b, c, d) = \begin{pmatrix} 1 & a & \psi_\lambda(a) & a\psi_\lambda(a) \\ 1 & b & \psi_\lambda(b) & a\psi_\lambda(b) \\ 1 & c & \psi_\lambda(c) & a\psi_\lambda(c) \\ 1 & d & \psi_\lambda(d) & a\psi_\lambda(d) \end{pmatrix}$.

Since no four distinct points in the curve $\mathcal{C} = \{(s, \psi(s), s\psi(s)) \mid s \in [0, 1]\}$ are coplanar, we have for all distinct $a, b, c, d \in [0, 1]$, with $0 \leq a < b < c < d \leq 1$, that the determinant of the matrix $M(a, b, c, d)$ is constant in sign:

$$\det(M(a, b, c, d)) > 0 \quad \text{or} \quad \det(M(a, b, c, d)) < 0.$$

This means that the determinant is either always positive or always negative.

Then we define $h(x) = q \det(M(0, p_1, x, 1)) + (1 - q) \det(M(0, p_2, x, 1))$. The signs of $h(x)$ on the intervals $[0, p_1]$ and $[p_2, 1]$ are opposite. According to the Intermediate Value Theorem Rudin et al. (1964), because $h(x)$ is continuous on $[0, 1]$ and $h(p_1) \times h(p_2) \leq 0$, there exists $p \in [p_1, p_2]$ satisfies $h(p) = 0$. Then $q v(p_1) + (1 - q) v(p_2), v(p), v(0), v(1)$ are coplanar. If there exists another $p' \neq p$ satisfies eq. (1), then $v(p), v(0), v(1), v(p')$ are coplanar, which is impossible. $\square$

**Lemma A.6.** *Let $0 < p_1 < p_2 < 1$ and $0 \leq q \leq 1$. Then there exist functions $p, x, y, z : \mathbb{R}^3 \to \mathbb{R}$ such that the following conditions hold:*

$$q v(p_1) + (1 - q) v(p_2) = x(p_1, p_2, q) v(p(p_1, p_2, q)) + y(p_1, p_2, q) v(0) + z(p_1, p_2, q) v(1),$$

*with $x(p_1, p_2, q) + y(p_1, p_2, q) + z(p_1, p_2, q) = 1$, that is,*

$$q v(p_1) + (1 - q) v(p_2), v(p(p_1, p_2, q)), v(0), v(1) \text{ are coplanar.}$$

*And the following inequalities hold:*

$$x(p_1, p_2, q) \geq 0, \quad y(p_1, p_2, q) \geq 0, \quad z(p_1, p_2, q) \geq 0.$$

*Proof of Lemma A.6.* We first prove that $x(p_1, p_2, q) \geq 0$.

To show $x(p_1, p_2, q) \geq 0$, we define $p_0(q) = p(p_1, p_2, q)$ and $x_0(q) = x(p_1, p_2, q)$. Consider the function:

$$h(s, q) = q \det(M(0, p_1, s, 1)) + (1 - q) \det(M(0, p_2, s, 1)).$$

Since $h(p_0(q), q) = 0$ and $h(s, q)$ is continuous on $[0, 1]^2$, we have: for any $q^*$

$$\lim_{q \to q^*} h(p_0(q), q) = h(\lim_{q \to q^*} p_0(q), q^*) = 0.$$

This implies $\lim_{q \to q^*} p_0(q) = p_0(q^*)$, which means $p_0$ is continuous.

Next, we can express $x_0(q)$ as:

$$x_0(q) = \frac{\det \begin{pmatrix} \boldsymbol{v}(0) \\ q\boldsymbol{v}(p_1) + (1-q)\boldsymbol{v}(p_2) \\ \boldsymbol{v}(1) \end{pmatrix}}{\det \begin{pmatrix} \boldsymbol{v}(0) \\ \boldsymbol{v}(p_0(q)) \\ \boldsymbol{v}(1) \end{pmatrix}}.$$

Since the determinant in the denominator is non-zero for distinct $p_1$ and $p_2$, $x_0(q)$ is continuous on $[0, 1]$. Given that $x_0(0) = 1$ and $x_0(1) = 1$, if $x_0(q) < 0$ for some $q \in (0, 1)$, by the Intermediate Value Theorem (Rudin et al., 1964), there exists $t$ such that $x_0(t) = 0$. This leads to $\boldsymbol{v}(0), \boldsymbol{v}(p_1), \boldsymbol{v}(p_2), \boldsymbol{v}(1)$ being coplanar, contradicting the condition.

Next, we will prove $y(p_1, p_2, q) \geq 0$.

When $q = 0$, we have $p(p_1, p_2, q) = p_2$, $x(p_1, p_2, q) = 1$, and $y(p_1, p_2, q) = z(p_1, p_2, q) = 0$. When $q = 1$, we have $p(p_1, p_2, q) = p_1$, $x(p_1, p_2, q) = 1$, and $y(p_1, p_2, q) = z(p_1, p_2, q) = 0$.

For $0 < q < 1$, we consider the case that $p_1 = 0$ in eq. (1), then $p = p_2$.

So we can define $p(0, p_2, q) = p_2$, $x(0, p_2, q) = 1 - q$, $y(0, p_2, q) = q$, $z(0, p_2, q) = 0$, and we have $\forall 0 \leq p_1 < p_2 < 1, 0 < q < 1, \exists p, x, y, z : \mathbb{R}^3 \to \mathbb{R}, \ s.t. \ q\boldsymbol{v}(p_1) + (1-q)\boldsymbol{v}(p_2) = x(p_1, p_2, q)\boldsymbol{v}(p(p_1, p_2, q)) + y(p_1, p_2, q)\boldsymbol{v}(0) + z(p_1, p_2, q)\boldsymbol{v}(1), x(p_1, p_2, q) + y(p_1, p_2, q) + z(p_1, p_2, q) = 1$

Then we fix $q$ and $p_2$ and define:

$$p_0(p_1) = p(p_1, p_2, q), \quad y_0(p_1) = y(p_1, p_2, q).$$

We then analyze the function:

$$h(s, p_1) = q \det(M(0, p_1, s, 1)) + (1-q) \det(M(0, p_2, s, 1)).$$

Since $h(p_0(p_1), p_1) = 0$ and $h(s, p_1)$ is continuous, we have:

$$\lim_{p_1 \to p_1'} h(p_0(p_1), p_1) = h(\lim_{p_1 \to p_1'} p_0(p_1), p_1') = 0.$$

This implies $\lim_{p_1 \to p_1'} p_0(p_1) = p_0(p_1')$, which means $p_0$ is continuous.

We can express $y_0(p_1)$ as:

$$y_0(p_1) = \frac{\det \begin{pmatrix} q\boldsymbol{v}(p_1) + (1-q)\boldsymbol{v}(p_2) \\ \boldsymbol{v}(p_0(p_1)) \\ \boldsymbol{v}(1) \end{pmatrix}}{\det \begin{pmatrix} \boldsymbol{v}(0) \\ \boldsymbol{v}(p_0(p_1)) \\ \boldsymbol{v}(1) \end{pmatrix}}.$$

Thus, $y_0(p_1)$ is continuous on $[0, 1)$. Given that $y_0(0) = q > 0$, if $y_0(p_1) < 0$ for some $p_1 \in (0, 1)$, then by the Intermediate Value Theorem (Rudin et al., 1964), there exists $t$ such that $y_0(t) = 0$. This leads to $\boldsymbol{v}(p), \boldsymbol{v}(p_1), \boldsymbol{v}(p_2), \boldsymbol{v}(1)$ being coplanar, contradicting the condition.

Finally, we will prove $z(p_1, p_2, q) \geq 0$.

When $q = 0$, we have $p(p_1, p_2, q) = p_2$, $x(p_1, p_2, q) = 1$, and $y(p_1, p_2, q) = z(p_1, p_2, q) = 0$. When $q = 1$, we have $p(p_1, p_2, q) = p_1$, $x(p_1, p_2, q) = 1$, and $y(p_1, p_2, q) = z(p_1, p_2, q) = 0$.

For $0 < q < 1$, we consider the case that $p_2 = 1$ in eq. (1). Then $p = p_1$.

So we can define $p(p_1, 1, q) = p_2$, $x(p_1, 1, q) = q$, $y(p_1, 1, q) = 0$, $z(p_1, 1, q) = 1 - q$ ,and we have $\forall 0 < p_1 < p_2 \leq 1, 0 < q < 1, \exists p, x, y, z : \mathbb{R}^3 \rightarrow \mathbb{R}, s.t. q\boldsymbol{v}(p_1) + (1 - q)\boldsymbol{v}(p_2) = x(p_1, p_2, q)\boldsymbol{v}(p(p_1, p_2, q)) + y(p_1, p_2, q)\boldsymbol{v}(0) + z(p_1, p_2, q)\boldsymbol{v}(1), x(p_1, p_2, q) + y(p_1, p_2, q) + z(p_1, p_2, q) = 1$

Then we fix $q$ and $p_1$ and define:

$$p_0(p_2) = p(p_1, p_2, q), \quad z_0(p_2) = z(p_1, p_2, q).$$

We analyze the function:

$$h(s, p_2) = q \det(M(0, p_1, s, 1)) + (1 - q) \det(M(0, p_2, s, 1)).$$

Since $h(p_0(p_2), p_2) = 0$ and $h(s, p_2)$ is continuous, we have:

$$\lim_{p_2 \to p_2'} h(p_0(p_2), p_2) = h(\lim_{p_2 \to p_2'} p_0(p_2), p_2') = 0.$$

This implies $\lim_{p_2 \to p_2'} p_0(p_2) = p_0(p_2')$, which means $p_0$ is continuous.

We can express $z_0(p_2)$ as:

$$z_0(p_2) = \frac{\det \begin{pmatrix} \boldsymbol{v}(0) \\ \boldsymbol{v}(p_0(p_2)) \\ q\boldsymbol{v}(p_1) + (1 - q)\boldsymbol{v}(p_2) \end{pmatrix}}{\det \begin{pmatrix} \boldsymbol{v}(0) \\ \boldsymbol{v}(p_0(p_2)) \\ \boldsymbol{v}(1) \end{pmatrix}}.$$

This shows that $z_0(p_2)$ is continuous on $[0, 1)$. Given that $z_0(0) = 1 - q > 0$, if $z_0(p_2) < 0$ for some $p_2 \in (0, 1)$, then by the Intermediate Value Theorem (Rudin et al., 1964), there exists $t$ such that $z_0(t) = 0$. This leads to $\boldsymbol{v}(p), \boldsymbol{v}(p_1), \boldsymbol{v}(p_2), \boldsymbol{v}(1)$ being coplanar, contradicting the condition.

Thus, we conclude that $x(p_1, p_2, q) \geq 0$, $y(p_1, p_2, q) \geq 0$, $z(p_1, p_2, q) \geq 0$. $\qquad\square$

Finally, we can prove that $\text{rep}(\theta, \psi_\lambda) = \hat{\theta}$, where $\theta \in \Theta^{3ciid}$ and $\hat{\theta} \in \hat{\Theta}_\lambda^{ciid}$, defines a bijection between the sets $\Theta^{3ciid}$ and $\hat{\Theta}_\lambda^{ciid}$. We first verify the surjectivity of the mapping:

$$\forall \hat{\theta} \in \hat{\Theta}_\lambda^{ciid}, \exists \theta \in \Theta^{3ciid} \text{ such that } \text{rep}(\theta, \psi) = \hat{\theta},$$

$$\iff \forall \theta^{ciid} \in \Theta^{ciid}, \exists \theta^{3ciid} \in \Theta^{3ciid} \text{ such that } \text{rep}(\theta^{3ciid}, \psi) = \text{rep}(\theta^{ciid}, \psi),$$

$$\iff \forall \theta^{ciid} \in \Theta^{ciid}, \exists \theta^{3ciid} \in \Theta^{3ciid} \text{ such that } \forall i \in [n],$$

$$\sum_{s \in \mathcal{S}_\theta} \Pr_{\theta^{ciid}}[S_i = s] \boldsymbol{v}(\Pr_{\theta^{ciid}}[\omega = 1 \mid S_i = s]) = \sum_{s \in \{0, p_i, 1\}} \Pr_{\theta^{3ciid}}[S_i = s] \boldsymbol{v}(\Pr_{\theta^{3ciid}}[\omega = 1 \mid S_i = s]).$$

In order to establish surjectivity, we first show that every signal set can be reduced to a canonical form, as captured in the following lemma.

**Lemma A.7** (Canonical Form Transformation). *Let*

$$\mathcal{S}_\theta = \{s^{(1)}, s^{(2)}, \ldots, s^{(m)}\}$$

*be a finite set of signals. Then there exists an iterative transformation that maps $\mathcal{S}_\theta$ to the canonical signal set $\{0, p, 1\}$ with*

$$\Pr_\theta[\omega = 1 \mid S_i = 0] = 0 \quad and \quad \Pr_\theta[\omega = 1 \mid S_i = 1] = 1.$$

*Moreover, this transformation preserves the equivalence of the corresponding point, that is,*

$$\sum_{j=1}^m \Pr[S_i = s^{(j)}] \boldsymbol{v}(\Pr[\omega = 1 \mid S_i = s^{(j)}]) = \sum_{s \in \{0, p, 1\}} \Pr[S_i = s] \boldsymbol{v}(s),$$

*Proof of Lemma A.7.* We prove the lemma by describing an iterative procedure that transforms $\mathcal{S}_\theta$ into the canonical form $\{0, p, 1\}$.

- **Step 1 (Group Identical Posteriors):**
  For any two signals $s^{(j)}$ and $s^{(k)}$ in $\mathcal{S}_\theta$ satisfying

  $$\Pr[\omega = 1 \mid S_i = s^{(j)}] = \Pr[\omega = 1 \mid S_i = s^{(k)}],$$

  Recall theorem A.1, we can group these signals by summing their marginal probabilities. Thus, without loss of generality, we may assume that every signal in $\mathcal{S}_\theta$ corresponds to a unique posterior probability.

- **Step 2 (Select and Replace a Pair of Intermediate Signals):**
  Choose two distinct signals $s^{(j)}$ and $s^{(k)}$ with posterior probabilities

  $$0 < \Pr[\omega = 1 \mid S_i = s^{(j)}] < \Pr[\omega = 1 \mid S_i = s^{(k)}] < 1.$$

  Define

  $$p_j := \Pr[\omega = 1 \mid S_i = s^{(j)}] \quad \text{and} \quad p_k := \Pr[\omega = 1 \mid S_i = s^{(k)}].$$

  By applying Lemma A.6, there exists a unique intermediate value $p^* \in [p_j, p_k]$ and nonnegative coefficients $x, y, z$ (satisfying $x + y + z = \Pr[S_i = s^{(j)}] + \Pr[S_i = s^{(k)}]$ ) such that

  $$\Pr[S_i = s^{(j)}] \, \boldsymbol{v}(p_j) + \Pr[S_i = s^{(k)}] \, \boldsymbol{v}(p_k) = x \, \boldsymbol{v}(p^*) + y \, \boldsymbol{v}(0) + z \, \boldsymbol{v}(1).$$

  In this step, the pair $\{s^{(j)}, s^{(k)}\}$ is replaced by a new composite signal corresponding to the intermediate value $p^*$, while the signals corresponding to 0 and 1 remain intact.

- **Step 3 (Iterate Until Canonical Form Is Achieved):**
  Replace the pair $\{s^{(j)}, s^{(k)}\}$ in $\mathcal{S}_\theta$ with the new set of signals $\{0, p^*, 1\}$ (noting that the signals corresponding to 0 and 1 may already exist in $\mathcal{S}_\theta$). Since each replacement reduces the number of signals with posterior probabilities strictly between 0 and 1, and because $\mathcal{S}_\theta$ is finite, this iterative process terminates after finitely many steps. The final signal set will therefore be in the canonical form

  $$\{0, p, 1\},$$

  where $p$ is the unique intermediate posterior (if any) between 0 and 1.

This completes the proof. $\qquad\square$

Lemma A.7 demonstrates that for every report structure $\hat{\theta} \in \hat{\Theta}_\lambda^{ciid}$ there exists an equivalent signal structure structure $\theta \in \Theta^{3ciid}$ satisfying

$$\mathrm{rep}(\theta, \psi_\lambda) = \hat{\theta}.$$

Thus, the mapping is surjective.

Next, we verify the injectivity of the mapping:

$$\forall \theta, \theta' \in \Theta^{3ciid}, \quad \mathrm{rep}(\theta, \psi) = \mathrm{rep}(\theta', \psi) \implies \theta = \theta'.$$

Since $\Pr_\theta[\omega = 1 \mid S_i = 0] = \Pr_{\theta'}[\omega = 1 \mid S_i = 0] = 0$ and $\Pr_\theta[\omega = 1 \mid S_i = 1] = \Pr_{\theta'}[\omega = 1 \mid S_i = 1] = 1$, we have

$$\text{rep}(\theta, \psi) = \text{rep}(\theta', \psi)$$

$$\iff \sum_{s_i \in \{0,p,1\}} \Pr_\theta[S_i = s_i]\boldsymbol{v}(\Pr_\theta[\omega = 1 \mid S_i = s_i]) = \sum_{s_i \in \{0,p,1\}} \Pr_{\theta'}[S_i = s_i]\boldsymbol{v}(\Pr_{\theta'}[\omega = 1 \mid S_i = s_i])$$

$$\iff \sum_{s_i \in \{0,p,1\}} \Pr_\theta[S_i = s_i]\boldsymbol{v}(\Pr_\theta[\omega = 1 \mid S_i = s_i]) - \sum_{s_i \in \{0,p,1\}} \Pr_{\theta'}[S_i = s_i]\boldsymbol{v}(\Pr_{\theta'}[\omega = 1 \mid S_i = s_i]) = 0$$

$$\iff (\Pr_\theta[S_i = 0] - \Pr_{\theta'}[S_i = 0])\boldsymbol{v}(0) + (\Pr_\theta[S_i = 1] - \Pr_{\theta'}[S_i = 1])\boldsymbol{v}(1)$$
$$+ \Pr_\theta[S_i = p]\boldsymbol{v}(\Pr_\theta[\omega = 1 \mid S_i = p]) - \Pr_{\theta'}[S_i = p]\boldsymbol{v}(\Pr_{\theta'}[\omega = 1 \mid S_i = p]) = 0$$

$$\iff \begin{cases} \Pr_\theta[S_i = 0] - \Pr_{\theta'}[S_i = 0] = 0 \\ \Pr_\theta[S_i = 1] - \Pr_{\theta'}[S_i = 1] = 0 \end{cases}$$
$$\wedge \left( \begin{cases} \boldsymbol{v}(\Pr_{\theta'}[\omega = 1 \mid S_i = p]) = \boldsymbol{v}(\Pr_{\theta'}[\omega = 1 \mid S_i = p]) \\ \Pr_\theta[S_i = p] - \Pr_{\theta'}[S_i = p] = 0 \end{cases} \vee \begin{cases} \Pr_{\theta'}[S_i = p] = 0 \\ \Pr_\theta[S_i = p] = 0 \end{cases} \right)$$

$$\iff \theta = \theta'$$

Together, these results confirm that $\hat{\theta} = \text{rep}(\theta, \psi_\lambda)$ is both injective and surjective, thereby defining a bijection between $\Theta^{3ciid}$ and $\hat{\Theta}_\lambda^{ciid}$.

Thus we proved

$$\text{rep}(\Theta^{3ciid}, \psi_\lambda) = \hat{\Theta}_\lambda^{ciid}$$

$\square$

## A.2 Optimal Robust Aggregator

We start to identify the optimal aggregator when experts exhibit bounded rationality. We prove that the simple majority voting is the optimal robust aggregator under certain conditions. In particular, for a given number of experts $n$, when the rationality level $\lambda$ is below a certain threshold $g(n)$, the majority voting $f^{maj}$ minimizes the regret. The threshold function $g(n)$ (for $n = 3$ to 20) visualized in fig. 4.

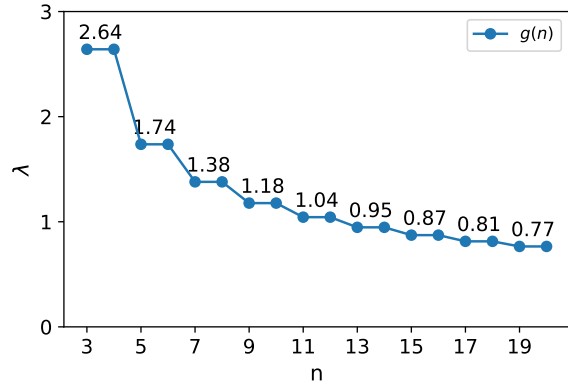

Figure 4: **Threshold $g(n)$ vs. Group Size $n$ ($n = 3$ to 20)** The plot begins at $n = 3$ because when $n \leq 2$, the majority voting is always the optimal robust aggregator, independent of $\lambda$. For $n \in [3, 20]$, $g(n)$ decreases with increasing $n$, with every even $n$ satisfying $g(n) = g(n-1)$. These properties indicate that as group size grows, the bounded rationality threshold becomes stricter, representing a lower rationality level to preserve the optimality of the majority voting rule.

**Intuition for the Step-Like Threshold $g(n)$:** As observed in fig. 4, the threshold function exhibits a step-like behavior where $g(2k) = g(2k - 1)$ for any integer $k \geq 2$. This occurs

because the expected utility of the majority vote aggregator $f^{maj}$ is mathematically identical for a group of $2k$ experts and a group of $2k-1$ experts. Specifically, for an odd group of $2k-1$ experts, no ties are possible. For an even group of $2k$ experts, ties are broken by a 0.5 random coin flip. The decision process for $2k$ experts is equivalent in expectation to randomly removing one expert and taking the majority vote of the remaining $2k-1$ experts. If the $2k$ experts are not in a tie, removing one expert does not change the majority outcome. If they are in a perfect $k$ vs. $k$ tie, removing one expert results in a $2k-1$ group whose majority vote will be '1' or '0' with 0.5 probability, matching the tie-breaking rule exactly. Since the expected utilities are identical, their optimal rationality thresholds $g(n)$ are also identical. Conversely, this symmetry breaks when moving from $2k$ to $2k+1$ (i.e., $g(2k+1) \neq g(2k)$), because a near-tie $(k+1, k)$ outcome is deterministic for $2k+1$ experts, but removing an expert can force a stochastic tie-breaking outcome in the $2k$ case, causing their expected utilities to diverge.

**Proof Sketch of Theorem 3.1: Optimal Robust Aggregator.**

The proof consists of two main steps:

1. **Pairwise Optimality of Majority Voting.**
   We aim to show $f^{maj}$ is the optimal solution to the following optimization problem under certain conditions:

   $$f^{maj} \in \arg\min_f \max_{\hat{\theta} \in \hat{\Theta}^{ciid}} R(f, \hat{\theta}).$$

   A direct approach to showing the worst-case regret optimality of the majority voting, $f^{maj}$, would involve maximizing over the entire space of possible report structures. To avoid this global maximization, we will employ approaches that consider each signal structure individually.

   One such approach would be to prove the stronger condition that $f^{maj}$ is universally optimal, meaning $f^{maj} \in \arg\min_f R(f, \hat{\theta})$ for all report structures $\hat{\theta} \in \hat{\Theta}^{ciid}$. While this would immediately imply worst-case regret optimality, $f^{maj}$ is not, in fact, universally optimal.

   Instead, we establish a weaker, yet sufficient, condition. We prove that, under certain conditions, for any pair of symmetric signal structures $\theta$ and $\theta'$ within the restricted set $\Theta^{3ciid}$, $f^{maj}$ minimizes the sum of expected regret:

   $$f^{maj} \in \arg\min_f \left( R(f, \text{rep}(\theta, \psi_\lambda)) + R(f, \text{rep}(\theta', \psi_\lambda)) \right).$$

   We then demonstrate how this pairwise optimality implies the desired worst-case regret optimality of $f^{maj}$, effectively avoiding the need to maximize over the entire space of signal structures.

2. **Regret Minimization Condition and Threshold Derivation.**
   First, we expand the formula $R(f, \text{rep}(\theta, \psi_\lambda)) + R(f, \text{rep}(\theta', \psi_\lambda))$ so that we can analyze each term of $f(x)$ separately. Then we proved $f^{maj}$ minimizes the sum of the expected regret if and only if

   $$\Pr_{\hat{\theta}} \left[ \omega = 1 \,\middle|\, X = \left\lfloor \frac{n-1}{2} \right\rfloor \right] \leq 0.5,$$

   where $X$ is the number of experts reporting 1. This further simplifies the question. Finally, we derive the threshold function $g(n)$ given the above condition. We prove that when $\lambda \leq g(n)$, the majority voting minimizes the sum of the expected regret under the pair of signal structures $\{\theta, \theta'\}$.

*Proof of Theorem 3.1: Optimal Robust Aggregator.*

**Pairwise Optimality of Majority Voting.**

Consider the following symmetric signal structures $\theta, \theta' \in \Theta^{3ciid}$ with $n$ experts

|  |  | $S_i = 0$ | $S_i = p$ | $S_i = 1$ |
|---|---|---|---|---|
| $\theta$ | $\omega = 0$ | $(1-\mu)(1-p_0)$ | $(1-\mu)p_0$ | $0$ |
|  | $\omega = 1$ | $0$ | $\mu p_1$ | $\mu(1-p_1)$ |
| $\theta'$ | $\omega = 0$ | $\mu(1-p_1)$ | $\mu p_1$ | $0$ |
|  | $\omega = 1$ | $0$ | $(1-\mu)p_0$ | $(1-\mu)(1-p_0)$ |

Let $p = \frac{\mu p_1}{\mu p_1 + (1-\mu)p_0}$. We define

$$\begin{cases} q_0 = \Pr[X_i = 1 \mid \omega = 0] = (1-p_0)\psi_\lambda(0) + p_0\psi_\lambda(p), \\ q_1 = \Pr[X_i = 1 \mid \omega = 1] = (1-p_1)\psi_\lambda(1) + p_1\psi_\lambda(p), \end{cases}$$

and we have the joint distribution of the two report structures

|  |  | $X_i = 0$ | $X_i = 1$ |
|---|---|---|---|
| $\hat{\theta}$ | $\omega = 0$ | $(1-\mu)(1-q_0)$ | $(1-\mu)q_0$ |
|  | $\omega = 1$ | $\mu(1-q_1)$ | $\mu q_1$ |
| $\hat{\theta}'$ | $\omega = 0$ | $\mu q_1$ | $\mu(1-q_1)$ |
|  | $\omega = 1$ | $(1-\mu)q_0$ | $(1-\mu)(1-q_0)$ |

Since the signal structures $\theta, \theta'$ are odd symmetric, we have

$$U(\text{opt}_{\hat{\theta}}, \hat{\theta}) = U(\text{opt}_{\hat{\theta}'}, \hat{\theta}').$$

Thus, to find the best response to $\{\theta, \theta'\}$, minimizing the regret is equivalent to maximizing the utility, i.e.,

$$\arg\min_f \max_{\hat{\theta} \in \{\theta, \theta'\}} R(f, \hat{\theta})$$
$$= \arg\min_f \max_{\hat{\theta} \in \{\theta, \theta'\}} U(\text{opt}_\theta, \hat{\theta}) - U(f, \hat{\theta})$$
$$= \arg\max_f \min_{\hat{\theta} \in \{\theta, \theta'\}} U(f, \hat{\theta})$$

Then, we need to prove $f^{maj}$ is the best response to $\{\theta, \theta'\}$, that is,

$$\min_{\theta \in \{\theta, \theta'\}} U(f^{maj}, \hat{\theta}) = \max_f \min_{\hat{\theta} \in \{\theta, \theta'\}} U(f, \hat{\theta}).$$

**Regret Minimization Condition and Threshold Derivation.**

We consider $\max_f (U(f, \hat{\theta}) + U(f, \hat{\theta}'))$.

$$U(f, \hat{\theta}) + U(f, \hat{\theta}') = \sum_{x=0}^{n} \left( (2f(x) - 1) \Pr_\theta[X = x, \omega = 1] + (1 - 2f(x)) \Pr_\theta[X = x, \omega = 0] \right)$$
$$+ \sum_{x=0}^{n} \left( (2f(x) - 1) \Pr_{\theta'}[X = x, \omega = 1] + (1 - 2f(x)) \Pr_{\theta'}[X = x, \omega = 0] \right)$$
$$= 1 + 2 \sum_{x=0}^{n} \binom{n}{x} f(x) \Big( (\mu q_1^x (1-q_1)^{n-x} + (1-\mu)q_0^{n-x}(1-q_0)^x$$
$$- (1-\mu)q_0^x(1-q_0)^{n-x} - \mu q_1^{n-x}(1-q_1)^x) \Big)$$

Next, we will prove that when $f^{maj}$ maximize the utility, majority voting is an optimal robust aggregator. Specially, since $\theta$ and $\theta'$ are odd symmetric, we have $U(f^{maj}, \hat{\theta}) = U(f^{maj}, \hat{\theta}')$. Thus we can prove

**Lemma A.8.** *If for any signal structure $\theta \in \Theta^{3ciid}$, the majority voting $f^{maj}$ satisfy*

$$f^{maj} \in \arg\max_f (U(f,\hat{\theta}) + U(f,\theta')).$$

*Then the majority voting $f^{maj}$ is a best response to $\Theta^{ciid}$, i.e.,*

$$f^{maj} \in \arg\max_f \min_{\theta \in \Theta^{ciid}} U(f,\hat{\theta}).$$

*Proof of Lemma A.8.*

$$\min_{\theta \in \{\theta,\theta'\}} U(f,\hat{\theta}) \leq \frac{U(f,\hat{\theta}) + U(f,\theta')}{2}$$

$$\implies \max_f \min_{\theta \in \{\theta,\theta'\}} U(f,\hat{\theta}) \leq \max_f \frac{(U(f,\hat{\theta}) + U(f,\theta'))}{2}$$

$$\implies \max_f \min_{\theta \in \{\theta,\theta'\}} U(f,\hat{\theta}) \leq \frac{U(f^{maj},\hat{\theta}) + U(f^{maj},\theta')}{2}$$

$$\implies \max_f \min_{\theta \in \{\theta,\theta'\}} U(f,\hat{\theta}) \leq \min_{\theta \in \{\theta,\theta'\}} U(f^{maj},\hat{\theta})$$

$$\implies \max_f \min_{\theta \in \{\theta,\theta'\}} U(f,\hat{\theta}) = \min_{\theta \in \{\theta,\theta'\}} U(f^{maj},\hat{\theta})$$

$$\implies U(\text{opt}_{\hat{\theta}},\hat{\theta}) - \max_f \min_{\theta \in \{\theta,\theta'\}} U(f,\hat{\theta}) = U(\text{opt}_{\hat{\theta}},\hat{\theta}) - \min_{\theta \in \{\theta,\theta'\}} U(f^{maj},\hat{\theta})$$

$$\implies \max_f \min_{\theta \in \{\theta,\theta'\}} R(f,\hat{\theta}) = \min_{\theta \in \{\theta,\theta'\}} R(f^{maj},\hat{\theta})$$

$$\implies f^{maj} \in \arg\min_f \max_{\theta \in \{\theta,\theta'\}} R(f,\hat{\theta})$$

Finally, we prove that when $\lambda < g(n)$, the best response to $\Theta^{ciid}$ is $f^{maj}$.

$$\forall \theta, \max_{\theta \in \Theta^{ciid}} R(f,\hat{\theta}) \geq \max_{\theta \in \{\theta,\theta'\}} R(f,\hat{\theta})$$

$$\implies \forall \theta, \min_f \max_{\theta \in \Theta^{ciid}} R(f,\hat{\theta}) \geq \min_f \max_{\theta \in \{\theta,\theta'\}} R(f,\hat{\theta})$$

$$\implies \forall \theta, \min_f \max_{\theta \in \Theta^{ciid}} R(f,\hat{\theta}) \geq R(f^{maj},\hat{\theta})$$

$$\implies \min_f \max_{\theta \in \Theta^{ciid}} R(f,\hat{\theta}) \leq \max_{\theta \in \Theta^{ciid}} R(f^{maj},\hat{\theta})$$

$$\implies \min_f \max_{\theta \in \Theta^{ciid}} R(f,\hat{\theta}) = \max_{\theta \in \Theta^{ciid}} R(f^{maj},\hat{\theta})$$

$$\implies f^{maj} \in \arg\max_f \min_{\theta \in \Theta^{ciid}} U(f,\hat{\theta})$$

$\square$

Let $f^* \in \arg\max_f (U(f,\hat{\theta}) + U(f,\theta'))$; $T(x) = \mu q_1^x (1-q_1)^{n-x} + (1-\mu)q_0^{n-x}(1-q_0)^x - (1-\mu)q_0^x(1-q_0)^{n-x} - \mu q_1^{n-x}(1-q_1)^x$; we have

$$f^*(x) = \begin{cases} 0, & T(x) < 0, \\ \text{any value in } [0,1], & T(x) = 0, \\ 1, & T(x) > 0. \end{cases}$$

Note that $\mu q_1^x(1-q_1)^{n-x} + (1-\mu)q_0^{n-x}(1-q_0)^x$ and $(1-\mu)q_0^x(1-q_0)^{n-x} + \mu q_1^{n-x}(1-q_1)^x$ are odd symmetric with respect to the midpoint $i = \frac{n}{2}$. Thus, we have

$$f^{maj} \in \arg\max_f (U(f,\hat{\theta}) + U(f,\hat{\theta}'))$$

if and only if for all integer $x < \frac{n}{2}$, the inequality

$$\mu q_1^x(1-q_1)^{n-x} + (1-\mu)q_0^{n-x}(1-q_0)^x \le (1-\mu)q_0^x(1-q_0)^{n-x} + \mu q_1^{n-x}(1-q_1)^x$$

holds.

To formalize this condition, we begin with the following lemma.

**Lemma A.9.** *If $q_0 \le q_1$, then $f^{maj} \in \arg\max_f(U(f,\hat{\theta}) + U(f,\hat{\theta}'))$ iff the following inequality holds:*

$$\mu(q_1(1-q_1))^{\lfloor \frac{n-1}{2} \rfloor}(1-2q_1) \le (1-\mu)(q_0(1-q_0))^{\lfloor \frac{n-1}{2} \rfloor}(1-2q_0).$$

*Proof of Lemma A.9.* To establish the equivalence between the condition in the lemma and the statement $f^{maj} \in \arg\max_f$, we need to prove the equivalence of

For all integer $i < \dfrac{n}{2}, \mu q_1^x(1-q_1)^{n-x} + (1-\mu)q_0^{n-x}(1-q_0)^x \le (1-\mu)q_0^x(1-q_0)^{n-x} + \mu q_1^{n-x}(1-q_1)^x$

and

$$\mu(q_1(1-q_1))^{\lfloor \frac{n-1}{2} \rfloor}(1-2q_1) \le (1-\mu)(q_0(1-q_0))^{\lfloor \frac{n-1}{2} \rfloor}(1-2q_0).$$

Let $I(n) = \begin{cases} 1, & n \text{ is odd} \\ 2, & n \text{ is even} \end{cases}$, then we have

$$\mu(q_1(1-q_1))^{\lfloor \frac{n-1}{2} \rfloor}(1-2q_1) \le (1-\mu)(q_0(1-q_0))^{\lfloor \frac{n-1}{2} \rfloor}(1-2q_0)$$
$$\iff \mu(q_1(1-q_1))^{\lfloor \frac{n-1}{2} \rfloor}((1-q_1)^{I(n)} - q_1^{I(n)}) \le (1-\mu)(q_0(1-q_0))^{\lfloor \frac{n-1}{2} \rfloor}((1-q_0)^{I(n)} - q_0^{I(n)})$$
$$\iff \mu q_1^{\lfloor \frac{n-1}{2} \rfloor}(1-q_1)^{\lceil \frac{n+1}{2} \rceil} + (1-\mu)q_0^{\lceil \frac{n+1}{2} \rceil}(1-q_0)^{\lfloor \frac{n-1}{2} \rfloor}$$
$$\le (1-\mu)q_0^{\lfloor \frac{n-1}{2} \rfloor}(1-q_0)^{\lceil \frac{n+1}{2} \rceil} - \mu q_1^{\lceil \frac{n+1}{2} \rceil}(1-q_1)^{\lfloor \frac{n-1}{2} \rfloor}$$
$$\iff \begin{cases} \mu q_1^{\frac{n}{2}}(1-q_1)^{\frac{n}{2}}(\frac{1-q_1}{q_1} - \frac{1-q_1}{q_1}) \le (1-\mu)q_0^{\frac{n}{2}}(1-q_0)^{\frac{n}{2}}(\frac{1-q_0}{q_0} - \frac{q_0}{1-q_0}) & n \text{ is even} \\ \mu q_1^{\frac{n}{2}}(1-q_1)^{\frac{n}{2}}(\sqrt{\frac{1-q_1}{q_1}} - \sqrt{\frac{1-q_1}{q_1}}) \le (1-\mu)q_0^{\frac{n}{2}}(1-q_0)^{\frac{n}{2}}(\sqrt{\frac{1-q_0}{q_0}} - \sqrt{\frac{q_0}{1-q_0}}) & n \text{ is odd} \end{cases}$$

When $i < \frac{n}{2}$, $0 < q_0 \le q_1 \le 0.5$, we define $t_0 = \frac{1-q_0}{q_0}$, $t_1 = \frac{1-q_1}{q_1}$, $s = \frac{n}{2} - i$. We know $1 \le t_1 \le t_0$. And

$$\mu q_1^x(1-q_1)^{n-x} + (1-\mu)q_0^{n-x}(1-q_0)^x \le (1-\mu)q_0^x(1-q_0)^{n-x} + \mu q_1^{n-x}(1-q_1)^x$$
$$\iff \mu q_1^{\frac{n}{2}}(1-q_1)^{\frac{n}{2}}(t_1^s - t_1^{-s}) \le (1-\mu)q_0^{\frac{n}{2}}(1-q_0)^{\frac{n}{2}}(t_0^s - t_0^{-s}) \qquad (5)$$

When $n$ is even, we know $0 < \mu q_1^{\frac{n}{2}}(1-q_1)^{\frac{n}{2}}(t_1 - t_1^{-1}) \le (1-\mu)q_0^{\frac{n}{2}}(1-q_0)^{\frac{n}{2}}(t_0 - t_0^{-1})$.

Since

$$\frac{t_1^s - t_1^{-s}}{t_1 - t_1^{-1}}$$

$$= \sum_{k=1}^{s} t_1^{2k-s-1}$$

$$= \frac{1}{2} \sum_{k=1}^{s} (t_1^{2k-s-1} + t_1^{s+1-2k})$$

$$\leq \frac{1}{2} \sum_{k=1}^{s} (t_0^{2k-s-1} + t_0^{s+1-2k})$$

$$= \sum_{k=1}^{s} t_0^{2k-s-1}$$

$$= \frac{t_0^s - t_0^{-s}}{t_0 - t_0^{-1}}$$

eq. (5) holds.

When $n$ is odd, we know $0 < \mu q_1^{\frac{n}{2}} (1-q_1)^{\frac{n}{2}} (t_1^{\frac{1}{2}} - t_1^{-\frac{1}{2}}) \leq (1-\mu) q_0^{\frac{n}{2}} (1-q_0)^{\frac{n}{2}} (t_0^{\frac{1}{2}} - t_0^{-\frac{1}{2}})$. Since

$$\frac{t_1^s - t_1^{-s}}{t_1^{\frac{1}{2}} - t_1^{-\frac{1}{2}}}$$

$$= \sum_{k=1}^{2s} t_1^{k-s-\frac{1}{2}}$$

$$= \frac{1}{2} \sum_{k=1}^{s} (t_1^{k-s-\frac{1}{2}} + t_1^{-k+s+\frac{1}{2}})$$

$$\leq \frac{1}{2} \sum_{k=1}^{s} (t_0^{k-s-\frac{1}{2}} + t_0^{-k+s+\frac{1}{2}})$$

$$= \sum_{k=1}^{s} t_0^{k-s-\frac{1}{2}}$$

$$= \frac{t_0^s - t_0^{-s}}{t_0^{\frac{1}{2}} - t_0^{-\frac{1}{2}}}$$

eq. (5) holds.

When $0.5 \leq q_0 \leq q_1 \leq 1$, we use $(\mu, q_0, q_1)$ to replace $(1-\mu, 1-q_1, 1-q_0)$. The equation is completely the same.

When $q_0 \leq 0.5 \leq q_1$, we have

$$\forall i < \frac{n}{2}, \begin{cases} \mu q_1^x (1-q_1)^{n-x} \leq \mu q_1^{n-x} (1-q_1)^x \\ (1-\mu) q_0^{n-x} (1-q_0)^x \leq (1-\mu) q_0^x (1-q_0)^{n-x} \end{cases}$$

$$\implies \forall i < \frac{n}{2}, \mu q_1^x (1-q_1)^{n-x} + (1-\mu) q_0^{n-x} (1-q_0)^x \leq (1-\mu) q_0^x (1-q_0)^{n-x} + \mu q_1^{n-x} (1-q_1)^x$$

And let $i = \lfloor \frac{n-1}{2} \rfloor$, we have

$$\mu q_1^x (1-q_1)^{n-x} + (1-\mu) q_0^{n-x} (1-q_0)^x \leq (1-\mu) q_0^x (1-q_0)^{n-x} + \mu q_1^{n-x} (1-q_1)^x$$

$$\implies \mu q_1^{\lfloor \frac{n-1}{2} \rfloor} (1-q_1)^{\lceil \frac{n+1}{2} \rceil} + (1-\mu) q_0^{\lceil \frac{n+1}{2} \rceil} (1-q_0)^{\lfloor \frac{n-1}{2} \rfloor}$$

$$\leq (1-\mu) q_0^{\lfloor \frac{n-1}{2} \rfloor} (1-q_0)^{\lceil \frac{n+1}{2} \rceil} - \mu q_1^{\lceil \frac{n+1}{2} \rceil} (1-q_1)^{\lfloor \frac{n-1}{2} \rfloor}$$

Thus, the condition always holds. □

The above lemma characterizes the condition under which the majority voting is the optimal robust aggregator in terms of the quantities $q_0$ and $q_1$. In order to connect these conditions with the parameter $\lambda$, we now provide a reduction result.

**Lemma A.10.** *Given $n, \lambda$, we have*

$$\forall \mu, p_0, p_1 \in [0,1], \ \mu(q_1(1-q_1))^{\lfloor \frac{n-1}{2} \rfloor}(1-2q_1) \le (1-\mu)(q_0(1-q_0))^{\lfloor \frac{n-1}{2} \rfloor}(1-2q_0)$$

$$\iff \forall q_0, q_1 \text{ such that } \psi_\lambda(0) < q_0 \le q_1 \le 0.5, \text{ the following inequality holds:}$$

$$\frac{(q_1(1-q_1))^{\lfloor \frac{n-1}{2} \rfloor}(1-2q_1)}{\psi_\lambda(1) - q_1} \le \frac{(q_0(1-q_0))^{\lfloor \frac{n-1}{2} \rfloor}(1-2q_0)}{\psi_\lambda(1) - q_0} \times \frac{2\lambda + \ln(1-q_0) - \ln(q_0)}{2\lambda - \ln(1-q_0) + \ln(q_0)}$$

*Proof of Lemma A.10.* Since $q_0 \le \psi_\lambda(p) \le q_1$, it follows that $q_0 > q_1$ is impossible.

Consider the case when $q_0 < 0.5 < q_1$. We have the following inequalities:

$$\begin{cases} \mu(q_1(1-q_1))^{\lfloor \frac{n-1}{2} \rfloor}(1-2q_1) \le 0 \le (1-\mu)(q_0(1-q_0))^{\lfloor \frac{n-1}{2} \rfloor}(1-2q_0), \\ \frac{(q_1(1-q_1))^{\lfloor \frac{n-1}{2} \rfloor}(1-2q_1)}{\psi_\lambda(1)-q_1} \le 0 \le \frac{(q_0(1-q_0))^{\lfloor \frac{n-1}{2} \rfloor}(1-2q_0)}{\psi_\lambda(1)-q_0} \times \frac{2\lambda+\ln(1-q_0)-\ln(q_0)}{2\lambda-\ln(1-q_0)+\ln(q_0)}. \end{cases}$$

Next, we consider the case when $q_0 \le q_1 \le 0.5$ or $0.5 \le q_0 \le q_1$. Define the following:

$$\mu' = 1 - \mu, \quad p'_0 = p_1, \quad p'_1 = p_0, \quad p' = \frac{\mu' p'_1}{\mu' p'_1 + (1-\mu')p'_0},$$

$$q'_0 = (1-p'_0)\psi_\lambda(0) + p'_0 \psi_\lambda(p'), \quad q'_1 = (1-p'_1)\psi_\lambda(1) + p'_1 \psi_\lambda(p').$$

We observe that $q'_0 = 1 - q_1$ and $q'_1 = 1 - q_0$. Thus, we have

$$\forall \mu, p_0, p_1 \in [0,1], \text{ if } 0.5 \le q_0 \le q_1 \le \psi_\lambda(1),$$
$$\mu(q_1(1-q_1))^{\lfloor \frac{n-1}{2} \rfloor}(1-2q_1) \le (1-\mu)(q_0(1-q_0))^{\lfloor \frac{n-1}{2} \rfloor}(1-2q_0)$$
$$\iff \forall q_0, q_1, \mu, p_0, p_1 \in [0,1], \text{ if } 0.5 \le q_0 \le q_1 \le \psi_\lambda(1),$$
$$\mu'(q'_1(1-q'_1))^{\lfloor \frac{n-1}{2} \rfloor}(1-2q'_1) \le (1-\mu')(q'_0(1-q'_0))^{\lfloor \frac{n-1}{2} \rfloor}(1-2q'_0)$$
$$\iff \forall \mu, p_0, p_1 \in [0,1], \text{ if } \psi_\lambda(0) \le q_0 \le q_1 \le 0.5,$$
$$\mu(q_1(1-q_1))^{\lfloor \frac{n-1}{2} \rfloor}(1-2q_1) \le (1-\mu)(q_0(1-q_0))^{\lfloor \frac{n-1}{2} \rfloor}(1-2q_0).$$

Thus, we conclude that

$$\forall \mu, p_0, p_1 \in [0,1], \ \mu(q_1(1-q_1))^{\lfloor \frac{n-1}{2} \rfloor}(1-2q_1) \le (1-\mu)(q_0(1-q_0))^{\lfloor \frac{n-1}{2} \rfloor}(1-2q_0)$$
$$\iff \forall \mu, p_0, p_1 \in [0,1], \text{ if } \psi_\lambda(0) \le q_0 \le q_1 \le 0.5,$$
$$\text{we have } \mu(q_1(1-q_1))^{\lfloor \frac{n-1}{2} \rfloor}(1-2q_1) \le (1-\mu)(q_0(1-q_0))^{\lfloor \frac{n-1}{2} \rfloor}(1-2q_0).$$

When $q_0 = \psi_\lambda(0)$, we have $\mu = 0$, $p_1 = 0$, or $p_0 = 0$.

If $\mu = 0$, then

$$\mu(q_1(1-q_1))^{\lfloor \frac{n-1}{2} \rfloor}(1-2q_1) = 0 \le (1-\mu)(q_0(1-q_0))^{\lfloor \frac{n-1}{2} \rfloor}(1-2q_0).$$

If $p_1 = 0$, then $q_1 = \psi_\lambda(1)$. Thus,

$$\mu(q_1(1-q_1))^{\lfloor \frac{n-1}{2} \rfloor}(1-2q_1) \le 0 \le (1-\mu)(q_0(1-q_0))^{\lfloor \frac{n-1}{2} \rfloor}(1-2q_0).$$

If $p_0 = 0$, then $p = 1$ and $q_1 = \psi_\lambda(1)$. Thus,

$$\mu(q_1(1-q_1))^{\lfloor \frac{n-1}{2} \rfloor}(1-2q_1) \le 0 \le (1-\mu)(q_0(1-q_0))^{\lfloor \frac{n-1}{2} \rfloor}(1-2q_0).$$

Therefore, when $q_0 = \psi_\lambda(0)$, we have

$$\mu(q_1(1-q_1))^{\lfloor \frac{n-1}{2} \rfloor}(1-2q_1) \le (1-\mu)(q_0(1-q_0))^{\lfloor \frac{n-1}{2} \rfloor}(1-2q_0).$$

Combining all cases, we conclude that

$$\forall \mu, p_0, p_1 \in [0,1], \ \mu(q_1(1-q_1))^{\lfloor \frac{n-1}{2} \rfloor}(1-2q_1) \leq (1-\mu)(q_0(1-q_0))^{\lfloor \frac{n-1}{2} \rfloor}(1-2q_0)$$
$$\Longleftrightarrow \forall \mu, p_0, p_1 \in [0,1], \ \text{if } \psi_\lambda(0) < q_0 \leq q_1 \leq 0.5,$$
$$\text{we have } \mu(q_1(1-q_1))^{\lfloor \frac{n-1}{2} \rfloor}(1-2q_1) \leq (1-\mu)(q_0(1-q_0))^{\lfloor \frac{n-1}{2} \rfloor}(1-2q_0).$$

Since $\psi_\lambda$ is strictly monotonically increasing on $[0,1]$, for $\psi_\lambda(0) < q_0 \leq q_1 \leq 0.5$, we know $p \in [\psi_\lambda^{-1}(q_0), \psi_\lambda^{-1}(q_1)]$, where $\psi_\lambda^{-1}$ is the inverse function of $\psi_\lambda$. For all $\psi_\lambda(0) < q_0 \leq q_1 \leq 0.5$ and $p \in [\psi_\lambda^{-1}(q_0), \psi_\lambda^{-1}(q_1)]$, let

$$p_1 = \frac{\psi_\lambda(1) - q_1}{\psi_\lambda(1) - \psi_\lambda(p)}, \quad p_0 = \frac{q_0 - \psi_\lambda(0)}{\psi_\lambda(p) - \psi_\lambda(0)}, \quad \mu = \frac{p_0 p}{p_0 p + p_1(1-p)}.$$

Then $\mu, p_0, p_1 \in [0,1]$ and

$$\begin{cases} p = \frac{\mu p_1}{\mu p_1 + (1-\mu)p_0}, \\ q_0 = (1-p_0)\psi_\lambda(0), + p_0 \psi_\lambda(p), \\ q_1 = (1-p_1)\psi_\lambda(1) + p_1 \psi_\lambda(p). \end{cases}$$

Therefore, we can conclude that:

$$\forall \mu, p_0, p_1 \in [0,1], \mu(q_1(1-q_1))^{\lfloor \frac{n-1}{2} \rfloor}(1-2q_1) \leq (1-\mu)(q_0(1-q_0))^{\lfloor \frac{n-1}{2} \rfloor}(1-2q_0)$$
$$\Longleftrightarrow \forall q_0, q_1 \text{ such that } \psi_\lambda(0) < q_0 \leq q_1 \leq 0.5, \text{ the following inequality holds}$$
$$\forall p \in [\psi_\lambda^{-1}(q_0), \psi_\lambda^{-1}(q_1)], \ \mu(q_1(1-q_1))^{\lfloor \frac{n-1}{2} \rfloor}(1-2q_1) \leq (1-\mu)(q_0(1-q_0))^{\lfloor \frac{n-1}{2} \rfloor}(1-2q_0).$$

Finally, we define $h(x) = \frac{e^{2\lambda x} - e^{-2\lambda x}}{x}$. Since $q_0 = (1-p_0)\psi_\lambda(0) + p_0\psi_\lambda(p)$ and $q_1 = (1-p_1)\psi_\lambda(1) + p_1\psi_\lambda(p)$, we have

$$p_0 = \frac{q_0 - \psi_\lambda(0)}{\psi_\lambda(p) - \psi_\lambda(0)}, \quad p_1 = \frac{\psi_\lambda(1) - q_1}{\psi_\lambda(1) - \psi_\lambda(p)}.$$

Then,

$$\frac{\frac{p}{1-p}}{}$$
$$= \frac{\mu p_1}{(1-\mu)p_0}$$
$$= \frac{\mu(\psi_\lambda(1) - q_1)(\psi_\lambda(p) - \psi_\lambda(0))}{(1-\mu)(q_0 - \psi_\lambda(0))(\psi_\lambda(1) - \psi_\lambda(p))}.$$

So,

$$\frac{\mu}{1-\mu}$$
$$= \frac{q_0 - \psi_\lambda(0)}{\psi_\lambda(1) - q_1} \times \frac{e^{2\lambda(1-p)} - e^{-2\lambda(1-p)}}{1-p} \times \frac{p}{e^{2\lambda p} - e^{-2\lambda p}}$$
$$= \frac{q_0 - \psi_\lambda(0)}{\psi_\lambda(1) - q_1} \times \frac{h(1-p)}{h(p)}.$$

Since $h(x) = \sum_{k=0}^{+\infty} \frac{2(2\lambda)^{2k+1} x^{2k}}{(2k+1)!}$, $h$ is strictly monotonically increasing on $[0, +\infty)$. Because $\psi_\lambda$ is strictly monotonically increasing on $[0, 1]$ and $q_0 \leq \psi_\lambda(p)$,

$$
\begin{aligned}
&\frac{\mu}{1-\mu} \\
=&\frac{q_0 - \psi_\lambda(0)}{\psi_\lambda(1) - q_1} \times \frac{h(1-p)}{h(p)} \\
\leq&\frac{q_0 - \psi_\lambda(0)}{\psi_\lambda(1) - q_1} \times \frac{h(1 - \psi_\lambda^{-1}(q_0))}{h(\psi_\lambda^{-1}(q_0))} \\
=&\frac{(\psi_\lambda(1) - q_0)(2\lambda - \ln(1 - q_0) + \ln q_0)}{(\psi_\lambda(1) - q_1)(2\lambda + \ln(1 - q_0) - \ln q_0)}.
\end{aligned}
$$

Especially, when $p = \psi_\lambda^{-1}(q_0)$,

$$
\frac{\mu}{1-\mu} = \frac{(\psi_\lambda(1) - q_0)(2\lambda - \ln(1 - q_0) + \ln q_0)}{(\psi_\lambda(1) - q_1)(2\lambda + \ln(1 - q_0) - \ln q_0)}.
$$

So,

$$
\forall \mu, p_0, p_1 \in [0, 1], \quad \mu(q_1(1 - q_1))^{\lfloor \frac{n-1}{2} \rfloor}(1 - 2q_1) \leq (1 - \mu)(q_0(1 - q_0))^{\lfloor \frac{n-1}{2} \rfloor}(1 - 2q_0)
$$

$\iff \forall q_0, q_1$ such that $\psi_\lambda(0) < q_0 \leq q_1 \leq 0.5$, the following inequality holds

$$
\forall p \in [\psi_\lambda^{-1}(q_0), \psi_\lambda^{-1}(q_1)], \ \mu(q_1(1 - q_1))^{\lfloor \frac{n-1}{2} \rfloor}(1 - 2q_1) \leq (1 - \mu)(q_0(1 - q_0))^{\lfloor \frac{n-1}{2} \rfloor}(1 - 2q_0)
$$

$\iff \forall q_0, q_1$ such that $\psi_\lambda(0) < q_0 \leq q_1 \leq 0.5$, the following inequality holds

$$
\frac{(q_1(1 - q_1))^{\lfloor \frac{n-1}{2} \rfloor}(1 - 2q_1)}{\psi_\lambda(1) - q_1} \leq \frac{(q_0(1 - q_0))^{\lfloor \frac{n-1}{2} \rfloor}(1 - 2q_0)}{\psi_\lambda(1) - q_0} \times \frac{2\lambda + \ln(1 - q_0) - \ln(q_0)}{2\lambda - \ln(1 - q_0) + \ln(q_0)}.
$$

$\square$

Lemma A.10 is crucial because it reformulates the condition from Lemma A.9 in terms of the parameter $\lambda$ and the transformed quantities $\psi_\lambda(0)$ and $\psi_\lambda(1)$. This reformulation paves the way for defining a threshold function $g(n)$ that governs the optimality of the majority voting.

Before generalizing to the case $n > 2$, we now examine the special scenario when $n \leq 2$.

**Lemma A.11.** *Given $n \leq 2$ and $\lambda$, if $q_0, q_1$ satisfy $\psi_\lambda(0) < q_0 \leq q_1 \leq \frac{1}{2}$, the following inequality holds:*

$$
\frac{(q_1(1 - q_1))^{\lfloor \frac{n-1}{2} \rfloor}(1 - 2q_1)}{\psi_\lambda(1) - q_1} \leq \frac{(q_0(1 - q_0))^{\lfloor \frac{n-1}{2} \rfloor}(1 - 2q_0)}{\psi_\lambda(1) - q_0} \times \frac{2\lambda + \ln(1 - q_0) - \ln(q_0)}{2\lambda - \ln(1 - q_0) + \ln(q_0)}
$$

*Proof of Lemma A.11.* Recall $h(x) = \sum_{k=0}^{+\infty} \frac{2(2\lambda)^{2k+1} x^{2k}}{(2k+1)!}$ in the proof of Lemma A.10, $h$ is strictly monotonically increasing on $[0, +\infty)$. Because $\psi_\lambda$ is strictly monotonically increasing

on $[0,1]$ and $q_0 \leq \psi_\lambda(p)$,

$$
\begin{aligned}
&\frac{\mu}{1-\mu} \\
=&\frac{q_0 - \psi_\lambda(0)}{\psi_\lambda(1) - q_1} \times \frac{h(1-p)}{h(p)} \\
\leq&\frac{q_0 - \psi_\lambda(0)}{\psi_\lambda(1) - q_1} \times \frac{h(1 - \psi_\lambda^{-1}(q_0))}{h(\psi_\lambda^{-1}(q_0))} \\
=&\frac{(\psi_\lambda(1) - q_0)(2\lambda - \ln(1-q_0) + \ln q_0)}{(\psi_\lambda(1) - q_1)(2\lambda + \ln(1-q_0) - \ln q_0)} \\
\leq&\frac{\psi_\lambda(1) - q_0}{\psi_\lambda(1) - q_1} \\
\leq&\frac{0.5 - q_0}{0.5 - q_1} \\
=&\frac{1 - 2q_0}{1 - 2q_1}
\end{aligned}
$$

Thus we proved $\mu(1 - 2q_1) \leq (1-\mu)(1 - 2q_0)$. $\qquad\square$

Lemma A.11 confirms that when there are at most two experts, the transformed inequality always holds. This simple case is important for understanding the behavior of the system in small dimensions. With these insights, we now turn to the more general case.

**Lemma A.12.** *Given $n > 2$ and $\lambda$, let*

$$
g(n) = \sup \left\{ \lambda \; \middle| \; \forall q_0, q_1 \; such \; that \; \psi_\lambda(0) \leq q_0 \leq q_1 \leq 0.5, \; the \; following \; inequality \; holds: \right.
$$

$$
\left. \frac{(q_1(1-q_1))^{\lfloor \frac{n-1}{2} \rfloor}(1 - 2q_1)}{\psi_\lambda(1) - q_1} \leq \frac{(q_0(1-q_0))^{\lfloor \frac{n-1}{2} \rfloor}(1 - 2q_0)}{\psi_\lambda(1) - q_0} \times \frac{2\lambda + \ln(1-q_0) - \ln(q_0)}{2\lambda - \ln(1-q_0) + \ln(q_0)} \right\}
$$

*Then, when $\lambda \leq g(n)$, if $q_0, q_1$ satisfy $\psi_\lambda(0) < q_0 \leq q_1 \leq \frac{1}{2}$, the following inequality holds:*

$$
\frac{(q_1(1-q_1))^{\lfloor \frac{n-1}{2} \rfloor}(1 - 2q_1)}{\psi_\lambda(1) - q_1} \leq \frac{(q_0(1-q_0))^{\lfloor \frac{n-1}{2} \rfloor}(1 - 2q_0)}{\psi_\lambda(1) - q_0} \times \frac{2\lambda + \ln(1-q_0) - \ln(q_0)}{2\lambda - \ln(1-q_0) + \ln(q_0)}
$$

*Proof of Lemma A.12.* Note that the term $\frac{\psi_\lambda(1) - q_1}{\psi_\lambda(1) - q_0} \cdot \frac{2\lambda + \ln(1-q_0) - \ln(q_0)}{2\lambda - \ln(1-q_0) + \ln(q_0)}$ is strictly monotonically decreasing in $\lambda$. Therefore, for any $\lambda_0 < \lambda_1 < g(n)$, if the inequality holds for $\lambda = \lambda_1$ with parameters $(n, \mu, q_0, q_1)$, it will also hold for $\lambda = \lambda_0$ with the same parameters.

$\qquad\square$

Lemma A.12 introduces the threshold function $g(n)$ for the general case when $n > 2$. This function serves as a critical cutoff: as long as $\lambda \leq g(n)$, the inequality (and hence the optimality conditions) established in the previous lemmas will hold.

Collecting the results from Lemmas A.9, A.10, A.11, A.12 and A.8, we conclude that when $\lambda < g(n)$ the majority voting $f^{maj}$ is indeed optimal. This completes the proof. $\qquad\square$

**Numerical Results.** To demonstrate the relationship between regret and rationality level $\lambda$, we conducted numerical experiments and calculated the regret of majority voting and optimal robust aggregator when the number of experts $n = 1, 3, 5$ and $\lambda$ is in $[0, 5]$, as shown in Figure 5. Regret is the maximum regret over all report structures induced by all c.i.i.d. signal structures. The optimal robust aggregator is computed using an online learning algorithm (Guo et al., 2023).

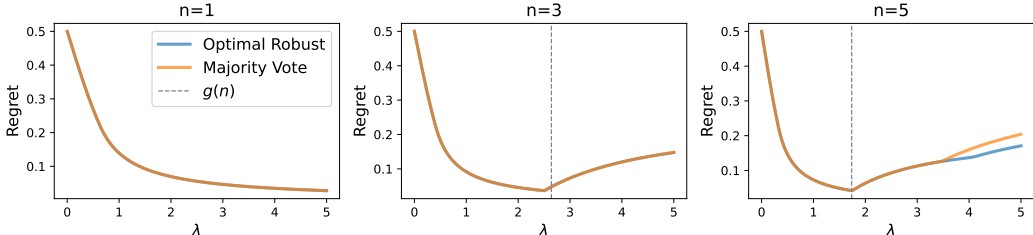

Figure 5: **Regret Comparison: Majority Voting vs. Optimal Robust Aggregator (Varying Rationality and Group Size).** These plots illustrate the relationship between the rationality level $\lambda$ (on the x-axis) and the regret (on the y-axis) for three different group sizes: $n = 1, 3,$ and $5$. Regret is the maximum regret over all report structures induced by all c.i.i.d. signal structures. The solid curves represent the regret incurred by the majority voting, $f^{maj}$, and the optimal robust aggregator, $\mathrm{opt}_{\hat{\Theta}^{ciid}}$. The dotted line indicates the threshold function $g(n)$. When the performance of majority voting matches that of the optimal robust aggregator (i.e., their regrets are equal), majority voting is optimal. With a single expert, any aggregation rule that follows that expert's decision, including majority voting, is trivially optimal. As $\lambda$ increases, reflecting increased rationality, the regret decreases. When $n \geq 3$ and $\lambda \leq g(n)$, majority voting remains optimal. However, the threshold $g(n)$ is not tight; majority voting may still be optimal even when $\lambda$ exceeds $g(n)$. Nevertheless, for $n = 5$ and sufficiently large $\lambda$, majority voting is no longer optimal. Furthermore, as $\lambda$ increases, the regret of both majority voting and the optimal robust aggregator initially decreases and then increases. This suggests that a moderate degree of bounded rationality can, in fact, improve aggregation performance.

### A.3 BOUNDED RATIONALITY ADVANTAGE

This subsection explores the impact of varying levels of rationality among experts on the aggregation outcomes. Specifically, we examine whether bounded rationality can sometimes lead to superior results compared to full rationality.

**Proof Sketch of Theorem 3.1: Bounded Rationality Advantage.**

The proof consists of three main steps:

1. **Single Expert Case ($n = 1$).**
   In the case of a single expert, the optimal aggregation rule is to follow the expert's decision regardless of rationality level $\lambda$. Hence, the overall utility of the aggregator is the same as the expected utility of the expert. Since a single expert attains the highest expected payoff when behaving in a fully rational manner, it follows that for any signal structure $\theta \in \Theta^{ciid}$, the maximal achievable utility is obtained under full rationality $\lambda \to \infty$.

2. **Two-Expert Case ($n = 2$).**
   When $n = 2$, we use the example mentioned in the introduction as the specific signal structure $\theta^*$. Under full rationality ($\lambda \to \infty$), experts always report the same value (here, $X_i = 0$) and the optimal aggregator yields a utility of 0.5. In contrast, when experts exhibit bounded rationality (with, say, $\lambda = 2.5$), their reporting behaviors differ, and we obtain

   $$U\left(\mathrm{opt}_{\mathrm{rep}(\theta^*, \psi_{\lambda=2.5})}, \ \mathrm{rep}\left(\theta^*, \psi_{\lambda=2.5}\right)\right) \approx 0.507674 > 0.5 = U\left(\mathrm{opt}_{\mathrm{rep}(\theta^*, \psi_{\lambda \to \infty})}, \ \mathrm{rep}\left(\theta^*, \psi_{\lambda \to \infty}\right)\right).$$

   This establishes that for $n = 2$ bounded rationality can lead to improved outcomes.

3. **Extension to $n > 2$ Experts.**
   Finally, we extend the argument to larger number of experts under $\theta^*$. Still, under full rationality ($\lambda \to \infty$), experts always report the same value and the optimal aggregator yields a utility of 0.5. For odd $n$, we prove that the majority voting yields a utility strictly greater than 0.5 when experts exhibit bounded rationality. For even

$n$, we show that the expected utility of majority voting remains the same as in the odd case $(n-1)$. Combining these two parts, we conclude that when $n \geq 2$, bounded rationality leads to superior aggregation performance compared to full rationality.

*Proof of Theorem 3.1: Bounded Rationality Advantage.*

**Single Expert Case $(n = 1)$.**

We begin by handling the $n = 1$ case.

**Lemma A.13.** *For $n = 1$, for any signal structure $\theta \in \Theta^{ciid}$ and any rationality level $\lambda$, it holds that*

$$\max_f U\left(f, rep\left(\theta, \psi_\lambda\right)\right) \leq U\left(f^{maj}, rep\left(\theta, \psi_{\lambda\to\infty}\right)\right).$$

*Proof of Lemma A.13.* For $n = 1$ the aggregator $f$ depends solely on the report $X_i$ of the single expert. For any signal structure $\theta$, we have

$$U\left(f, \operatorname{rep}\left(\theta, \psi_\lambda\right)\right) = \mathbb{E}_{\operatorname{rep}(\theta,\psi_\lambda)}\, u(f(X_i), \omega)$$

$$= \sum_{s_i \in \mathcal{S}_\theta} \left(2 \Pr_\theta[f(X_i) = 1 \mid S_i = s_i] - 1\right)\left(2 \Pr_\theta[\omega = 1 \mid S_i = s_i] - 1\right) \Pr_\theta[S_i = s_i]$$

$$\leq \sum_{s_i \in \mathcal{S}_\theta} \left|2 \Pr_\theta[\omega = 1 \mid S_i = s_i] - 1\right| \Pr_\theta[S_i = s_i].$$

Recall that the response function of a rational expert is

$$\psi_{\lambda\to\infty}(p) = \begin{cases} 1, & p > \frac{1}{2}, \\ 0.5, & p = \frac{1}{2}, \\ 0, & p < \frac{1}{2}, \end{cases}$$

and the majority voting is given by

$$f^{maj}(x) = x.$$

Thus, the utility of $f^{maj}$ under rational is

$$U\left(f^{maj}, \operatorname{rep}\left(\theta, \psi_{\lambda\to\infty}\right)\right)$$

$$= \sum_{s_i \in \mathcal{S}_\theta} \left(2 \psi_{\lambda\to\infty}\left(\Pr_\theta[\omega = 1 \mid S_i = s_i]\right) - 1\right)\left(2 \Pr_\theta[\omega = 1 \mid S_i = s_i] - 1\right) \Pr_\theta[S_i = s_i]$$

$$= \sum_{s_i \in \mathcal{S}_\theta} \left|2 \Pr_\theta[\omega = 1 \mid S_i = s_i] - 1\right| \Pr_\theta[S_i = s_i].$$

This immediately yields the desired inequality. $\square$

**Two-Expert Case $(n = 2)$.**

Next, we consider the case $n = 2$. Define the signal structure $\theta^*$ with the following joint distribution:

| | | $S_i = 0$ | $S_i = 1$ |
|---|---|---|---|
| $\theta^*$ | $\omega = 0$ | $\frac{3}{8}$ | $\frac{3}{8}$ |
| | $\omega = 1$ | $0$ | $\frac{1}{4}$ |

so that

$$\Pr[\omega = 1 \mid S_i = 0] = 0, \quad \Pr[\omega = 1 \mid S_i = 1] = \frac{2}{5}.$$

Under absolute rationality $(\lambda \to \infty)$, the experts always report $X_i = 0$ and the optimal aggregator (which, in this example, is the majority voting) yields

$$U\left(\operatorname{opt}_\infty, \operatorname{rep}(\theta^*, \psi_{\lambda\to\infty})\right) = 0.5.$$

When experts are bounded rational with $\lambda^* = 5$, the report distributions differ. In the case $n = 2$, letting

$$q_0 = \Pr[X_i = 1 \mid \omega = 0] = \frac{1}{2 + 2e^5} + \frac{1}{2 + 2e}, \quad q_1 = \Pr[X_i = 1 \mid \omega = 1] = \frac{1}{1 + e},$$

and choosing the aggregator

$$\mathrm{opt}_{\lambda^*}(x) = \begin{cases} 0, & x \leq 1, \\ 1, & x = 2, \end{cases}$$

one obtains

$$U\Big(\mathrm{opt}_{\lambda^*}, \mathrm{rep}(\theta^*, \psi_{\lambda=2.5})\Big) \approx 0.507674 > U\big(\mathrm{opt}_\infty, \mathrm{rep}(\theta^*, \psi_{\lambda\to\infty})\big).$$

This shows that for $n = 2$,

$$\max_f U\left(f, \mathrm{rep}\left(\theta^*, \psi_{\lambda=2.5}\right)\right) > \max_f U\left(f, \mathrm{rep}\left(\theta^*, \psi_{\lambda\to\infty}\right)\right).$$

**Extension to $n > 2$ Experts.**

Finally, we consider the case when $n > 2$. When $n$ is odd, the following lemma holds

**Lemma A.14.** *When $n$ is odd, we have*

$$U\left(f^{maj}, rep(\theta^*, \psi_{\lambda=\lambda^*})\right) > 0.5$$

*Proof.*

$$U\left(f^{maj}, \mathrm{rep}(\theta^*, \psi_{\lambda=\lambda^*})\right) = \sum_{x=0}^n \left(2f^{maj}(x) - 1\right) \left(\Pr[\omega = 1, X = x] - \Pr[\omega = 0, X = x]\right)$$

$$= \sum_{x=0}^{\lfloor \frac{n-1}{2} \rfloor} \left(\Pr[\omega = 0] \left(\Pr[X = x \mid \omega = 0] - \Pr[X = n - x \mid \omega = 0]\right)\right.$$
$$\left. - \Pr[\omega = 1] \left(\Pr[X = x \mid \omega = 1] - \Pr[X = n - x \mid \omega = 1]\right)\right)$$

$$= \sum_{x=0}^{\lfloor \frac{n-1}{2} \rfloor} \binom{n}{x} \left(0.75 \left(q_0^x(1 - q_0)^{n-x} - q_0^{n-x}(1 - q_0)^x\right)\right.$$
$$\left. - 0.25 \left(q_1^x(1 - q_1)^{n-x} - q_1^{n-x}(1 - q_1)^x\right)\right)$$

$$= 0.75 \left(\sum_{x=0}^{\lfloor \frac{n-1}{2} \rfloor} \binom{n}{x} q_0^x(1 - q_0)^{n-x} - \sum_{x=0}^{\lfloor \frac{n-1}{2} \rfloor} \binom{n}{x} q_0^{n-x}(1 - q_0)^x\right)$$

$$- 0.25 \left(\sum_{x=0}^{\lfloor \frac{n-1}{2} \rfloor} \binom{n}{x} q_1^x(1 - q_1)^{n-x} - \sum_{x=0}^{\lfloor \frac{n-1}{2} \rfloor} \binom{n}{x} q_1^{n-x}(1 - q_1)^x\right)$$

$$= 0.5 + 0.25 \left(\sum_{x=0}^{\lceil \frac{n-1}{2} \rceil} \binom{n}{x} q_1^{n-x}(1 - q_1)^x + \sum_{x=0}^{\lfloor \frac{n-1}{2} \rfloor} \binom{n}{x} q_1^{n-x}(1 - q_1)^x\right)$$

$$- 0.75 \left(\sum_{x=0}^{\lceil \frac{n-1}{2} \rceil} \binom{n}{x} q_0^{n-x}(1 - q_0)^x + \sum_{x=0}^{\lfloor \frac{n-1}{2} \rfloor} \binom{n}{x} q_0^{n-x}(1 - q_0)^x\right).$$

When $n$ is odd, we have $\lceil \frac{n-1}{2} \rceil = \lfloor \frac{n}{2} \rfloor = \frac{n-1}{2}$, and the expression becomes:

$$U\left(f^{maj}, \text{rep}(\theta^*, \psi_{\lambda=\lambda^*})\right) = 0.5 + \sum_{x=0}^{\frac{n-1}{2}} \binom{n}{x} \left(0.5 q_1^{n-x}(1-q_1)^x - 1.5 q_0^{n-x}(1-q_0)^x\right)$$

$$= 0.5 + \sum_{x=0}^{\frac{n-1}{2}} \binom{n}{x} \left(0.5 q_1^2(1-q_1)\left(q_1^{n-2x-1}(1-q_1)^{x-1}\right)\right.$$

$$- 1.5 q_0^2(1-q_0)\left(q_0^{n-2x-1}(1-q_0)^{x-1}\right))$$

$$\geq 0.5 + \sum_{x=0}^{\frac{n-1}{2}} \binom{n}{x} \left(0.5 q_1^2(1-q_1) - 1.5 q_0^2(1-q_0)\right) q_0^{n-x-1}(1-q_0)^{x-1}$$

$$> 0.5. \qquad\qquad (\text{since } 0.5 q_1^2(1-q_1) - 1.5 q_0^2(1-q_0) > 0)$$

$\square$

And when $n$ is even, the expected utility of the majority voting is equal to the case when there are $n-1$ experts:

**Lemma A.15.** *Let $n$ be even, and let $\hat{\theta}_{n-1} \in \hat{\Theta}^{ciid}$ denote the report structure with $n-1$ experts, and $\hat{\theta}_n \in \hat{\Theta}^{ciid}$ denote the report structure with $n$ experts. If the joint distribution of $\hat{\theta}_{n-1}$ and $\hat{\theta}_n$ for a certain expert is identical, then the utility of the majority voting remains the same. Formally, we have:*

$$\forall x_i, w \in \{0,1\}, \Pr_{\hat{\theta}_{n-1}}[X_i = x_i, \omega = w] = \Pr_{\hat{\theta}_n}[X_i = x_i, \omega = w]$$

$$\implies U(f^{maj}, \hat{\theta}_{n-1}) = U(f^{maj}, \hat{\theta}_n)$$

Since $n-1$ is odd, and Lemma A.14 have already proven that the utility is greater than 0.5 when the number of experts is odd, it follows that the utility is also greater than 0.5 when $n$ is even.

*Proof of Lemma A.15.* Let

$$\begin{cases} \mu = \Pr_{\hat{\theta}_n}[\omega = 1], q_0 = \Pr_{\hat{\theta}_n}[X_i = 1 \mid \omega = 0], \\ q_1 = \Pr_{\hat{\theta}_n}[X_i = 1 \mid \omega = 1]. \end{cases}$$

And we have

$$U(f^{maj}, \hat{\theta}_n)$$

$$= \sum_{x=0}^{n} \left(2f^{maj}(x) - 1\right) \left(\Pr_{\hat{\theta}_n}[\omega = 1, X = x] - \Pr_{\hat{\theta}_n}[\omega = 0, X = x]\right)$$

$$= \sum_{x=0}^{\lfloor \frac{n-1}{2} \rfloor} \left(\Pr_{\hat{\theta}_n}[\omega = 0] \left(\Pr_{\hat{\theta}_n}[X = x \mid \omega = 0] - \Pr_{\hat{\theta}_n}[X = n - x \mid \omega = 0]\right)\right.$$

$$\left. - \Pr_{\hat{\theta}_n}[\omega = 1] \left(\Pr_{\hat{\theta}_n}[X = x \mid \omega = 1] - \Pr_{\hat{\theta}_n}[X = n - x \mid \omega = 1]\right)\right)$$

$$= 1 - 2\mu + \mu \left(2 \sum_{x=0}^{\frac{n}{2}-1} \binom{n}{x} q_1^{n-x}(1-q_1)^x + \binom{n}{\frac{n}{2}} q_1^{\frac{n}{2}}(1-q_1)^{\frac{n}{2}}\right)$$

$$- (1-\mu) \left(2 \sum_{x=0}^{\frac{n}{2}-1} \binom{n}{x} q_0^{n-x}(1-q_0)^x + \binom{n}{\frac{n}{2}} q_0^{\frac{n}{2}}(1-q_0)^{\frac{n}{2}}\right)$$

$$= 1 - 2\mu + \mu \left(2q_1^n + 2 \sum_{x=1}^{\frac{n}{2}-1} \left(\binom{n-1}{x} + \binom{n-1}{x-1}\right) q_1^{n-x}(1-q_1)^x + 2\binom{n-1}{\frac{n}{2}-1} q_1^{\frac{n}{2}}(1-q_1)^{\frac{n}{2}}\right)$$

$$- (1-\mu) \left(2q_0^n + 2 \sum_{x=1}^{\frac{n}{2}-1} \left(\binom{n-1}{x} + \binom{n-1}{x-1}\right) q_0^{n-x}(1-q_0)^x + 2\binom{n-1}{\frac{n}{2}-1} q_0^{\frac{n}{2}}(1-q_0)^{\frac{n}{2}}\right)$$

$$= 1 - 2\mu + \mu \left(2 \sum_{x=0}^{\frac{n}{2}-1} \binom{n-1}{x} q_1^{n-x}(1-q_1)^x + 2 \sum_{x=1}^{\frac{n}{2}} \binom{n-1}{x-1} q_1^{n-x}(1-q_1)^x\right)$$

$$- (1-\mu) \left(2 \sum_{x=0}^{\frac{n}{2}-1} \binom{n-1}{x} q_0^{n-x}(1-q_0)^x + 2 \sum_{x=1}^{\frac{n}{2}} \binom{n-1}{x-1} q_0^{n-x}(1-q_0)^x\right)$$

$$= 1 - 2\mu + \left(2 \sum_{x=0}^{\frac{n}{2}-1} \binom{n-1}{x} \left(\mu q_1^{n-x-1}(1-q_1)^x - (1-\mu)q_0^{n-x-1}(1-q_0)^x\right)\right)$$

$$= \sum_{x=0}^{\lfloor \frac{n-2}{2} \rfloor} \left(\Pr_{\hat{\theta}_{n-1}}[\omega = 0] \left(\Pr_{\hat{\theta}_{n-1}}[X = x \mid \omega = 0] - \Pr_{\hat{\theta}_{n-1}}[X = n - 1 - x \mid \omega = 0]\right)\right.$$

$$\left. - \Pr_{\hat{\theta}_{n-1}}[\omega = 1] \left(\Pr_{\hat{\theta}_{n-1}}[X = x \mid \omega = 1] - \Pr_{\hat{\theta}_{n-1}}[X = n - 1 - x \mid \omega = 1]\right)\right)$$

$$= U(f^{maj}, \hat{\theta}_{n-1})$$

$$\square$$

Thus, we have shown that when $n > 2$,

$$U(f^{maj}, \text{rep}(\theta^*, \psi_{\lambda=\lambda^*})) > \max_f U(f, \text{rep}(\theta^*, \psi_{\lambda \to \infty})).$$

$$\square$$

**Numerical Results.** To further substantiate our theoretical findings, we performed numerical simulations to assess the performance of two aggregation methods on report structure $\text{rep}(\theta^*, \psi_\lambda)$: the majority voting ($f^{maj}$) and the omniscient aggregator ($\text{opt}_{\text{rep}(\theta^*, \psi_\lambda)}$). These simulations explore how their utility varies with the rationality level $\lambda$ and the number of

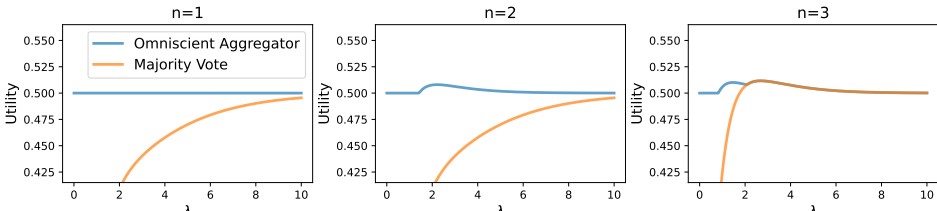

Figure 6: **Utility Comparison: Majority Voting vs. Omniscient Aggregator (Varying Rationality and Group Size)** The utility comparison is performed under the specific signal structure $\theta^*$, illustrated in the introduction. Under full rationality ($\lambda \to \infty$), the maximum achievable utility is 0.5. For $n = 1$, the omniscient aggregator maintains a constant utility of 0.5, while the majority voting converges to 0.5 as $\lambda \to \infty$. For $n = 2$, the omniscient aggregator achieves a utility greater than 0.5 for certain values of $\lambda$. Majority voting yields the same utility for $n = 1$ and $n = 2$. However, this omniscient aggregator requires knowledge of the signal structure, whereas the majority voting does not. For $n = 3$, the benefits of bounded rationality become more apparent. The majority voting, which requires no knowledge of the signal structure, can exceed a utility of 0.5, reaching its peak performance at intermediate values of $\lambda$.

experts $n$. The results are summarized in fig. 6. These numerical results support our theoretical conclusions. In particular, under the specific signal structure $\theta^*$ and when $n \geq 2$, they demonstrate that bounded rationality can enhance the decision-making performance, enabling the aggregators to surpass the maximum utility of 0.5 attainable under full rationality.

## B  DETAILED STUDY SETUP FOR SECTION 4

### B.1  LICENSES AND TERMS OF USE

**MathQA Dataset**  The MathQA dataset (Amini et al., 2019) is publicly available under the Apache License 2.0. It can be accessed at `https://huggingface.co/datasets/allenai/math_qa`.

**GPT-4o-mini Model**  The GPT-4o-mini-2024-07-18 model used in this study is provided by OpenAI. Its usage is governed by OpenAI's Terms of Use and Service Terms, available at `https://openai.com/policies/terms-of-use` and `https://openai.com/policies/service-terms/`, respectively.

### B.2  BAYESIAN DECISION-MAKING STUDY

**Study Design**  The Bayesian Decision-Making Study is framed as follows: There are two boxes, the left box ($\omega = 1$) and the right box ($\omega = 0$), each containing red and blue balls in specific proportions. In the right box ($\omega = 0$), the proportion of red balls is $\Pr[S_i = r \mid \omega = 0]$, and the proportion of blue balls is $\Pr[S_i = b \mid \omega = 0]$. Similarly, in the left box ($\omega = 1$), the corresponding proportions are $\Pr[S_i = r \mid \omega = 1]$ and $\Pr[S_i = b \mid \omega = 1]$. A box is selected randomly with a prior probability $\mu = \Pr[\omega = 1]$ of choosing the left box. A ball is then drawn from the chosen box, and its observed color, $S_i \in \{r, b\}$, represents the signal, where $S_i = r$ denotes a red ball and $S_i = b$ a blue ball. The task is to infer which box was selected based on the observed signal $S_i$, the provided prior $\Pr[\omega = 1]$ and the conditional probabilities $\Pr[S_i \mid \omega]$.

Because the number of scenarios we can query is finite, the probabilities are discretized as follows: the prior probability $\Pr[\omega = 1]$, the probability of observing a red ball given the left box $\Pr[S_i = r \mid \omega = 1]$, and the probability of observing a red ball given the right box $\Pr[S_i = r \mid \omega = 0]$ are each expressed as fractions with $N = 5$ as the denominator. Specifically, these probabilities take values from the set $\{0/N, 1/N, 2/N, \ldots, N/N\}$. This discretization creates a set of decision-making tasks that span a range of prior and conditional

probabilities. After removing the illegal and impossible cases, we have a total of 400 possible discretized scenarios.

The study utilizes the *gpt-4o-mini* model as the expert decision-maker. The model is queried under three distinct temperature settings $\{0, 0.5, 1\}$, each representing different levels of rationality. A **Chain of Thought** (CoT) prompting strategy is employed, requiring the model to explain its reasoning before delivering a binary decision ("L" for the left box, $X_i = 1$, or "R" for the right box, $X_i = 0$). To capture variability in the model's stochastic behavior, each decision-making scenario is repeated 20 times for each temperature setting. The *decision proportion* for given scenario and temperature is defined as the empirical probability of choosing the left box ("L") across all repetitions.

| Temperature | Coefficients ($\lambda$) | Standard Errors ($SE$) | z-values ($z$) | p-values ($p$) |
|---|---|---|---|---|
| 0.0 | $\infty$ | - | - | - |
| 0.5 | 26.49 | 10.23 | 2.59 | <0.001 |
| 1.0 | 17.86 | 5.44 | 3.28 | <0.001 |

Table 1: **Logistic regression results for different temperature settings.** The regression coefficient $\lambda$, corresponding to the rationality level in the quantal response: higher $\lambda$ values indicate more deterministic behavior, while lower values suggest increased randomness. For the case with temperature 0.0, the observed perfect determinism in choices leads to complete separation in the logistic regression, resulting in theoretically infinite coefficients. The standard error ($SE$) reflects the precision of $\lambda$, with smaller values indicating greater accuracy. z-values and p-values confirm the statistical significance of the findings.

### B.3   Multiple-Choice Question Answering Study

**Dataset Description**   The MathQA dataset (Amini et al., 2019) is a collection of complex mathematical reasoning problems presented as multiple-choice questions. Each problem consists of a problem statement and a set of five options $(A, B, C, D, E)$, with one correct answer. The dataset is designed to test logical reasoning, arithmetic, algebra, and other mathematical concepts, making it a challenging benchmark for evaluating LLMs. For this study, we use a subset of 5000 MathQA problems to ensure computational feasibility, focusing on tasks that require probabilistic reasoning and decision-making.

**Study Design**   To evaluate LLM performance, we deploy the following experimental setup. The LLM is queried under three distinct temperature settings ($t = 0.0, 0.5, 1.0$) to simulate varying levels of rationality. For each question, the model generates 20 responses per temperature setting. Because the questions offer more than two options, we use plurality voting ($f^{plu}$), a generalization of majority voting. This involves selecting the option with the highest frequency among the sampled responses. In the event of a tie, the aggregator randomly chooses among the tied options.

**Bootstrapping**   To assess the robustness of aggregation, we perform bootstrapping by randomly sampling $n$ responses (without replacement) from the 20 generated responses per question. This bootstrapping procedure is repeated 1000 times for each combination of temperature setting $t$ and expert group size $n = 1, 3, 5$. Accuracy is defined as the proportion of aggregated answers that match the ground truth, with error bars representing the standard error of the mean across bootstrap iterations.

**An Example**   We present an example in our study of how increasing temperature (i.e., adding randomness) counterintuitively increases expected accuracy. In our study, we ask gpt-4o-mini the following MathQA question.

"An alloy of copper and zinc contains copper and zinc in the ratio 3:5. Another alloy contains copper and zinc in the ratio 6:2. In what ratio should the two alloys be mixed to achieve equal proportions of copper and zinc?" The options are "1 : 2", "2 : 2", "2 : 5", "2 : 6", "2 : 7".

At $t = 0.0$ (deterministic mode), after querying LLM, 19/20 responses selecting the incorrect option B ("2 : 2"). Consequently, plurality voting always selects the incorrect option B. However, at $t = 1.0$, 4/20 responses correctly identify A ("1 : 2") Despite most responses being incorrect (16/20 chose B), the increased variance allows the aggregation to sometimes identify the correct answer.

One possible explanation is ambiguity in the question regarding the order of the mixing ratio (first alloy to second, or second to first). In deterministic mode ($t = 0.0$), the LLM frequently calculates the correct 2:1 ratio (first to second), but then fails to select the ground truth answer of 1:2. Some responses even explicitly justify this error, stating, "However, since the options provided do not include 2:1, we will look for the closest match. The closest option to 2:1 is **B) 2:2**, which represents equal proportions.".

However, the increased randomness in stochastic mode ($t = 1.0$) allows the LLM to exhibit greater "intelligence." Some responses cleverly recognize the intended meaning, acknowledging the discrepancy and stating, "None of the options list a ratio of 2:1 directly. There is no exact match, but the closest option that can relate to 2 is option A: 1:2 if we consider it in context of inverse." This suggests that the added randomness enables the LLM to explore a wider range of interpretations and, in some cases, deduce the correct answer despite the question's ambiguity. See the Section C.2 for more details.

## C  PROMPTS FOR LLMS

This section provides detailed descriptions of the prompts employed in our studies with Large Language Models (LLMs). These prompts are used to assess various aspects of hallucination detection, including prior answer generation, semantic clustering, and posterior answer generation. Our intention is to provide transparency and reproducibility for our method and those we compare against, particularly in terms of how LLMs interact with different prompt structures.

These prompts were developed specifically for use in studies described in the main body of the paper. While they have been designed to provide effective results, we note that they may still be open to refinement for further improvements in performance and reliability.

## C.1  BAYESIAN DECISION-MAKING STUDY

---

**User Prompt**

```
### Scenario:
You are given two boxes of balls. Each box contains 100 balls,
    which can be either red or blue. A box is selected at random,
     and one ball is drawn from the selected box. However, the
    identity of the selected box is not revealed.

You are provided with the following information:
- Left Box contains {left_red} red balls and {left_blue} blue
    balls.
- Right Box contains {right_red} red balls and {right_blue} blue
     balls.
- The probability of selecting the Left Box is {left_probability
    }%.
- The probability of selecting the Right Box is {
    right_probability}%.

### Question:
If the ball drawn is {color}, which box has been selected?

Please first briefly show the calculations in the <reason></
    reason> section, and then provide your final answer in the <
    answer></answer> section.
Your answer in <answer></answer> should **strictly** be a single
     uppercase letter: "L" for the Left Box or "R" for the Right
    Box. No additional text, no explanations.
Your response must contain <answer></answer> section with a
    definitive answer of either "L" or "R" in it.

Example response:
<reason># Briefly show your calculations here, using the
    provided information.</reason>
<answer># A single uppercase letter: L or R</answer>
```

Example Input

```
### Scenario:
You are given two boxes of balls. Each box contains 100 balls,
    which can be either red or blue. A box is selected at random,
     and one ball is drawn from the selected box. However, the
    identity of the selected box is not revealed.

You are provided with the following information:
- Left Box contains 100 red balls and 0 blue balls.
- Right Box contains 50 red balls and 50 blue balls.
- The probability of selecting the Left Box is 20.0%.
- The probability of selecting the Right Box is 80.0%.

### Question:
If the ball drawn is red, which box has been selected?

Please first briefly show the calculations in the <reason></
    reason> section, and then provide your final answer in the <
    answer></answer> section.
Your answer in <answer></answer> should **strictly** be a single
     uppercase letter: "L" for the Left Box or "R" for the Right
    Box. No additional text, no explanations.
Your response must contain <answer></answer> section with a
    definitive answer of either "L" or "R" in it.

Example response:
<reason># Briefly show your calculations here, using the
    provided information.</reason>
<answer># A single uppercase letter: L or R</answer>
```

---

**Example Output**

```
<reason>To find the probability of selecting each box given that
     a red ball is drawn, we can use Bayes' theorem.

Let:
- \( P(L) = 0.2 \) (probability of selecting the Left Box)
- \( P(R) = 0.8 \) (probability of selecting the Right Box)
- \( P(\text{Red} | L) = 1 \) (probability of drawing a red ball
     from the Left Box)
- \( P(\text{Red} | R) = 0.5 \) (probability of drawing a red
    ball from the Right Box)

Using the law of total probability, we calculate \( P(\text{Red
    }) \):
\[
P(\text{Red}) = P(\text{Red} | L) \cdot P(L) + P(\text{Red} | R)
     \cdot P(R) = 1 \cdot 0.2 + 0.5 \cdot 0.8 = 0.2 + 0.4 = 0.6
\]

Now, we can find \( P(L | \text{Red}) \) using Bayes' theorem:
\[
P(L | \text{Red}) = \frac{P(\text{Red} | L) \cdot P(L)}{P(\text{
    Red})} = \frac{1 \cdot 0.2}{0.6} = \frac{0.2}{0.6} = \frac
    {1}{3}
\]

Similarly, we find \( P(R | \text{Red}) \):
\[
P(R | \text{Red}) = \frac{P(\text{Red} | R) \cdot P(R)}{P(\text{
    Red})} = \frac{0.5 \cdot 0.8}{0.6} = \frac{0.4}{0.6} = \frac
    {2}{3}
\]

Since \( P(R | \text{Red}) > P(L | \text{Red}) \), it is more
    likely that the Right Box was selected if a red ball is drawn
    .</reason>
<answer>R</answer>
```

## C.2    MULTIPLE-CHOICE QUESTION ANSWERING STUDY

We first provide the template of prompt, then we provide the example input and two example outputs. The input and outputs correspond to the example in Section B.3.

---

**User Prompt of Math**

```
### Scenario:
Please review the following math question carefully:

### Question:
{question_str}

### Options:
{options_str}

### Instructions:
1. Provide your step-by-step calculations using your math
   knowledge, enclosed within <reason></reason> tags.
2. Select the best answer from the options provided, and place
   the uppercase letter of your final answer within <answer></
   answer> tags.

Please ensure your reasoning clearly supports the selected
   answer and that your final answer is one of the provided
   options. If none of the options are precisely correct, choose
    the closest one.

Example response:
<reason># Your reasoning and calculations go here</reason>
<answer># A single uppercase letter: the option you choose</
   answer>
```

---

Example Input

```
### Scenario:
Please review the following math question carefully:

### Question:
an alloy of copper and zinc contains copper and zinc in the
    ratio 3 : 5 . another alloy of copper and zinc contains
    copper and zinc in the ratio 6 : 2 . in what ratio should the
     two alloys be mixed so that the resultant alloy contains
    equal proportions of copper and zinc ?

### Options:
A) 1 : 2
B) 2 : 2
C) 2 : 5
D) 2 : 6
E) 2 : 7

### Instructions:
1. Provide your step-by-step calculations using your math
    knowledge, enclosed within <reason></reason> tags.
2. Select the best answer from the options provided, and place
    the uppercase letter of your final answer within <answer></
    answer> tags.

Please ensure your reasoning clearly supports the selected
    answer and that your final answer is one of the provided
    options. If none of the options are precisely correct, choose
     the closest one.

Example response:
<reason># Your reasoning and calculations go here</reason>
<answer># A single uppercase letter: the option you choose</
    answer>
```

---

**Example Output When t Equals 0**

```
<reason>
Let's denote the first alloy (with a copper to zinc ratio of
    3:5) as Alloy A and the second alloy (with a copper to zinc
    ratio of 6:2) as Alloy B.

1. **Determine the composition of Alloy A:**
   - The total parts in Alloy A = 3 (copper) + 5 (zinc) = 8
      parts.
   - Copper in Alloy A = 3/8 of the total weight.
   - Zinc in Alloy A = 5/8 of the total weight.

2. **Determine the composition of Alloy B:**
   - The total parts in Alloy B = 6 (copper) + 2 (zinc) = 8
      parts.
   - Copper in Alloy B = 6/8 = 3/4 of the total weight.
   - Zinc in Alloy B = 2/8 = 1/4 of the total weight.

3. **Let the weights of Alloy A and Alloy B be x and y
    respectively.**
   - The total copper from both alloys = (3/8)x + (3/4)y.
   - The total zinc from both alloys = (5/8)x + (1/4)y.

4. **Set the equation for equal proportions of copper and zinc
    :**
   \[
   (3/8)x + (3/4)y = (5/8)x + (1/4)y
   \]

5. **Clear the fractions by multiplying through by 8:**
   \[
   3x + 6y = 5x + 2y
   \]

6. **Rearranging gives:**
   \[
   3x - 5x + 6y - 2y = 0 \implies -2x + 4y = 0 \implies 2x = 4y
      \implies x/y = 4/2 = 2/1
   \]

7. **Thus, the ratio of Alloy A to Alloy B is 2:1.**
   - This means we need to mix Alloy A and Alloy B in the ratio
      of 2:1.

Since the question asks for the ratio of the two alloys, we can
    express this as 2:2 (which simplifies to 1:1) or 2:1. However
    , since the options provided do not include 2:1, we will look
     for the closest match.

The closest option to 2:1 is **B) 2:2**, which represents equal
    proportions.

Therefore, the final answer is:
</reason>
<answer>B</answer>
```

---

**Example Output When t Equals 1**

```
<reason>To solve the problem, we need to first determine the
    composition of each alloy in terms of copper and zinc.

1. **First Alloy (Ratio 3:5)**:
   - The total parts in the first alloy = 3 + 5 = 8 parts.
   - Copper in the first alloy = (3/8) * 100% = 37.5%
   - Zinc in the first alloy = (5/8) * 100% = 62.5%

2. **Second Alloy (Ratio 6:2)**:
   - The total parts in the second alloy = 6 + 2 = 8 parts.
   - Copper in the second alloy = (6/8) * 100% = 75%
   - Zinc in the second alloy = (2/8) * 100% = 25%

Let's denote the first alloy as A (copper 37.5%, zinc 62.5%) and
    the second alloy as B (copper 75%, zinc 25%). We want to mix
     these two alloys in the ratio x:y such that the resultant
    alloy has equal proportions of copper and zinc.

We will express the copper and zinc contents in the resultant
    mixture:

- Copper from A = \(x \times 0.375\)
- Zinc from A = \(x \times 0.625\)

- Copper from B = \(y \times 0.75\)
- Zinc from B = \(y \times 0.25\)

The total copper content in the mixture (A + B) is:
\[
\text{Total Copper} = 0.375x + 0.75y
\]

The total zinc content in the mixture (A + B) is:
\[
\text{Total Zinc} = 0.625x + 0.25y
\]

We want these two quantities to be equal:
\[
0.375x + 0.75y = 0.625x + 0.25y
\]

Rearranging this equation, we get:
\[
0.375x - 0.625x + 0.75y - 0.25y = 0
\]
\[
-0.25x + 0.5y = 0
\]
\[
0.5y = 0.25x
\]
```

### Example Output When t Equals 1

```
\[
y = 0.5x
\]

So, the ratio of \(x : y\) can be written as \(x : 0.5x\) or
    simplifying gives:
\[
x : y = 2 : 1
\]

This means we need to mix the alloys in a ratio of 2:1 (2 parts
    of A for every 1 part of B).

Now, we should see which option matches this finding:
- None of the options list a ratio of 2:1 directly. There is no
    exact match, but the closest option that can relate to 2 is
    option A: 1:2 if we consider it in context of inverse.

Since the required ratio of the alloys results in copper and
    zinc being equal, let's interpret the problem in inverse
    terms as it signifies that for choosing A and B:
To maintain a balance of ratios, the best option that reflects
    appropriate mixing is **option A** which provides a guideline
     for consideration, perhaps in taking more from the second
    one (due to its higher concentration of copper).

So, based on this reasoning:
</reason>
<answer>A</answer>
```

