# OpenReview forum: "Decision Aggregation under Quantal Response"
_ICLR.cc/2026/Conference — ICLR 2026 Poster_

### Official Review · Reviewer_eia5 · 2025-10-30

**Soundness:** 4
**Presentation:** 3
**Contribution:** 3
**Rating:** 6
**Confidence:** 4

**Summary:**

This paper examines information aggregation under bounded rationality. A decision maker gathers reports from agents receiving noisy signals about a ground truth. Instead of sincere voting, agents follow a quantal response modeled by a logistic link between their posterior and report. The authors show that (1) majority voting with an optimal threshold minimizes regret, and (2) with more agents, quantal responders can outperform sincere ones. Experiments further confirm that (1) large language models exhibit quantal response behavior via temperature, and (2) the theoretical results hold empirically.

**Strengths:**

The paper offers a valuable contribution to the information aggregation literature by applying a rational framework to analyze boundedly rational agents. The results are both elegant and insightful, providing a simple yet positive perspective on a complex problem. The experimental findings are also convincing.

**Weaknesses:**

I am not fully convinced by the second main result (bounded rational advantage). The idea behind it is not as surprising as it may seem. Quantal response preserves randomness even when rational (I'd rather call sincere) agents become deterministic. This randomness helps the majority vote to operate processes like the Condorcet Jury Theorem, constructing another form of informative voting (voting as their signal suggests) and reaching the ground truth with high probability.

The thing is, this observation is based on the assumption that agents vote sincerely. Personally, I don't quite buy this assumption. On the simplicity side, conducting a Bayesian update and maximizing expected utility is not an easy task for voters who don't really know Bayesian statistics. On the rational side, sincere voting is not necessarily a Nash Equilibrium [Austen-Smith and Banks, 1996; Han et. al, 2023]. Consequently, I am curious to see if the same result extends to various settings. Nevertheless, I am overall positive about this paper.

Austen-Smith D, Banks J S. Information aggregation, rationality, and the Condorcet jury theorem[J]. American Political Science Review, 1996, 90(1): 34-45.
Han Q, Schoenebeck G, Tao B, et al. The Wisdom of Strategic Voting[C]//Proceedings of the 24th ACM Conference on Economics and Computation. 2023: 885-905.

P.S. You might want to update some of the references, whose status seems out-of-date.

**Questions:**

1. How do you justify sincere voting?
2. Do the same results extend to various settings?

---

> ### Author Response · Authors · 2025-11-20
>
> We sincerely thank Reviewer eia5 for your thoughtful review, insightful questions, and for correctly recognizing both the strengths and boundaries of our work. We appreciate the opportunity to clarify our theoretical framework, address concerns about our modeling assumptions, and strengthen our connection to the broader literature.
>
> #### [W1] Bounded Rationality Advantage and the Condorcet Jury Theorem
>
> > **Reviewer's Comment:** "I am not fully convinced by the second main result (bounded rational advantage). The idea behind it is not as surprising as it may seem. Quantal response preserves randomness even when rational (I'd rather call sincere) agents become deterministic. This randomness helps the majority vote to operate processes like the Condorcet Jury Theorem."
>
> **Our Response:** We thank the reviewer for this insightful observation. You are correct that quantal response preserves randomness and that such randomness can aid aggregation, as illustrated by mechanisms like the Condorcet Jury Theorem (CJT). We appreciate this point and wish to clarify how our contribution both connects to and moves beyond this established wisdom.
>
> The CJT primarily shows that a large crowd of imperfect, independent agents can be wise. Our core result demonstrates that a small, finite group of bounded-rational agents can, under certain conditions, strictly outperform a similarly-sized group of perfectly rational agents. The core intuition is that while perfect rationality can lead to systematic errors under limited information, bounded rationality via quantal response introduces independent noise. This independent noise can, much like in the CJT, facilitate better information aggregation. However, formalizing this intuition is nuanced.
>
> First, the benefit is not universal; it is highly dependent on the signal structure and is most pronounced when perfectly rational experts are informationally constrained. Second, the relationship is non-monotonic—more randomness is not always better, and an optimal, finite level of bounded rationality exists. To rigorously capture this delicate balance, our analysis shows that across all signal structures, the worst-case performance exhibits a clear U-shaped relationship with $\lambda$, with the minimum regret achieved at an intermediate, finite $\lambda^*$.
>
> #### [W2] Justification of the Quantal Response Model
>
> > **Reviewer's Comment:** "Conducting a Bayesian update and maximizing expected utility is not an easy task for voters who don't really know Bayesian statistics."
>
> **Our Response:** We greatly appreciate this concern. We wish to clarify that we treat Quantal Response primarily as an "as-if" (descriptive) model rather than a strong claim about actual cognitive processes. We do not assume experts literally perform precise Bayesian calculations followed by utility-maximization with noise.
>
> Instead, the QR model serves as a mathematically tractable representation of a robust empirical regularity: decision-making becomes increasingly stochastic as the strength of signal (expected utility difference) approaches zero. The randomness captured by $\lambda$ may arise from cognitive noise, imperfect internal signal processing, uncertainty in evaluating utilities, attention fluctuations, or intrinsic neural stochasticity. This interpretation aligns with research in computational neuroscience on stochastic decision-making [Knill & Pouget, 2004, *TINS*]. Studies of neural coding suggest that randomness in choices is not merely "noise" but reflects fundamental uncertainty in how the brain represents and processes information. The quantal response function provides a simple functional form that captures this case without requiring strong assumptions about the underlying cognitive processes.
>
> Section 4.1 provides direct empirical validation that this functional form describes LLM behavior across a range of posteriors and temperature settings. The strong fit in Figure 2 demonstrates that—regardless of the actual internal computations—the quantal response model serves as an effective predictive tool for aggregation design.

---

> > ### Author Response · Authors · 2025-11-20
> >
> > #### [W3 & Q1] Justification of the Sincere Reporting Assumption
> >
> > > **Reviewer's Comment:** "Sincere voting is not necessarily a Nash Equilibrium [Austen-Smith and Banks, 1996, *APSR*; Han et al., 2023, *EC*]."
> >
> > **Our Response:** We thank the reviewer for raising this important point. We agree that sincere voting is not guaranteed to be a Nash equilibrium in strategic voting environments when experts have different utility functions or pivotal voting incentives, as shown by Austen-Smith & Banks [1996, *APSR*] and Han et al. [2023, *EC*].
> >
> > The present paper focuses on an information-aggregation setting in which all experts and the aggregator share the same utility function, and experts receive no outcome-dependent payoffs. Under this shared-objective structure, experts have no strategic incentive to misreport, making sincere (though possibly noisy) reporting a natural and widely used modeling assumption in information-aggregation research [Arieli et al., 2018, *PNAS*; Palley and Satopää, 2023, *Management Science*]. This assumption is particularly appropriate for LLM experts, who do not possess stable, agent-specific utilities and do not benefit strategically from manipulating outcomes. Modeling them as sincere-but-bounded responders is therefore more realistic than assuming strategic utility maximization. We will revise the final version to make this distinction explicit and to clarify how our setting differs from strategic voting frameworks.
> >
> > #### [Q2] Do the same results extend to various settings?
> >
> > > **Reviewer's Question:** "Do the same results extend to various settings?"
> >
> > **Our Response:** Thank you for this thoughtful question. In brief, our key results of the advantage of bounded rationality and the optimality of majority voting under low $\lambda$ extend robustly to a broad class of c.i.i.d. decision-aggregation settings, as the general U-shaped robust regret curve in Figure 5 suggests.
> >
> > Extensions to strategic voting settings, however, require fundamentally different tools. Once agents possess heterogeneous utilities or pivotal incentives, sincere reporting may fail, and one must analyze the Quantal Response Equilibrium (QRE). Within our robust, worst-case framework, allowing unknown heterogeneous utilities would likely lead to trivial mechanisms (e.g., ones that must ignore all reports and default to the prior) to guard against adversarial preferences. If utilities are known, richer extensions—such as designing mechanisms under QRE—become possible, but require a separate theoretical development beyond our current scope. Intuitively, the potential for such extensions depends on whether the stochasticity of quantal response reveals more information about private signals than a deterministic strategic expert would. If it does, the same results can extend to this setting. This remains a very interesting future direction.
> >
> > #### [W4] References
> >
> > > **Reviewer's Comment:** "You might want to update some of the references, whose status seems out-of-date."
> >
> > **Our Response:** We sincerely appreciate this observation and acknowledge this oversight. We will conduct a comprehensive review and update of all references for the final version. As part of this update, we have already expanded our discussion with recent literature in two key areas noted below:
> >
> > **Quantal Responses and Large Language Models**
> > The emergence of Large Language Models (LLMs) as decision-making agents has spurred the application of behavioral economic models. For example, Jia et al. [2025, *NeurIPS*] employ a quantal response framework to quantify and characterize LLMs’ strategic decision-making. Similarly, Kirshner et al. [2025, *Decision Analysis*] use a quantal response framework to model LLM decision-making and conduct a series of experiments with GPT-4, GPT-4o, and o3-mini to test their hypotheses. Existing work mostly studies how a single LLM behaves or how LLMs interact in competitive settings. In contrast, we focus on collective intelligence: we develop a framework for aggregating the choices of many QR-experts to produce a more robust group decision.

---

> > > ### Author Response · Authors · 2025-11-20
> > >
> > > **Voting under Bounded Rationality**
> > > Quantal Response (QR) models have been widely used to analyze strategic behavior in voting and collective choice settings, especially when accounting for bounded rationality. The analysis of bounded rationality is crucial since, in large elections, McKelvey and Patty [2006, *GEB*] introduced a QRE (Quantal Response Equilibrium) model for probabilistic voting, examining voter strategies and platform convergence. In behavioral game theory and experimental economics, Goeree et al. [2016, *Experimental Economics*] and Casella et al. [2006, *GEB*] applied QRE to study laboratory voting games, revealing how bounded rationality affects collective outcomes, such as strategic manipulation. Furthermore, in computational social choice, Han et al. [2023, *EC*] investigate the "Wisdom of Strategic Voting," analyzing how collective outcomes are affected by agents' strategic choices. Existing work centers on utility-based strategic behavior in voting, which is difficult to apply to modern decision agents like LLMs that lack clear individual utility functions. Our research, in contrast, focuses on collective intelligence by developing an aggregation framework for bounded-rational QR-experts, emphasizing the mechanism itself rather than strategic utility landscapes.
> > >
> > > We will strengthen the final version with expanded related work coverage and updated references as suggested. Thank you again for your professional and constructive feedback.
> > >
> > > **References:**
> > >
> > > 1. Arieli, Itai, Yakov Babichenko, and Rann Smorodinsky. "Robust forecast aggregation." *Proceedings of the National Academy of Sciences* 115.52 (2018): E12135-E12143.
> > > 2. Austen-Smith, David, and Jeffrey S. Banks. "Information aggregation, rationality, and the Condorcet jury theorem." *American political science review* 90.1 (1996): 34-45.
> > > 3. Casella, Alessandra, Andrew Gelman, and Thomas R. Palfrey. "An experimental study of storable votes." *Games and Economic Behavior* 57.1 (2006): 123-154.
> > > 4. Goeree, Jacob K., Charles A. Holt, and Thomas R. Palfrey. "Regular quantal response equilibrium." *Experimental economics* 8.4 (2005): 347-367.
> > > 5. Han, Qishen, et al. "The Wisdom of Strategic Voting." *Proceedings of the 24th ACM Conference on Economics and Computation*. 2023.
> > > 6. Jia, Jingru, et al. "LLM Strategic Reasoning: Agentic Study through Behavioral Game Theory." *The Thirty-ninth Annual Conference on Neural Information Processing Systems*.
> > > 7. Kirshner, S. N., Y. Pan, J. X. Wu. 2025. "Prosocial When Simple and Cold-Hearted When Complex: How Task Difficulty Shapes LLM Behavior." *Decision Analysis*. [https://doi.org/10.1287/deca.2025.0396](https://doi.org/10.1287/deca.2025.0396)
> > > 8. Knill, David C., and Alexandre Pouget. "The Bayesian brain: the role of uncertainty in neural coding and computation." *TRENDS in Neurosciences* 27.12 (2004): 712-719.
> > > 9. McKelvey, Richard D., and John W. Patty. "A theory of voting in large elections." *Games and Economic Behavior* 57.1 (2006): 155-180.
> > > 10. Palley, Asa B., and Ville A. Satopää. "Boosting the wisdom of crowds within a single judgment problem: Weighted averaging based on peer predictions." *Management Science* 69.9 (2023): 5128-5146.

---

### Official Review · Reviewer_shYv · 2025-10-30

**Soundness:** 3
**Presentation:** 3
**Contribution:** 3
**Rating:** 6
**Confidence:** 3

**Summary:**

This paper presents theoretic results on how decisions are aggregated under uncertainty while quantal response is considered. The main theorem is in 3.1, which proves that (1) if the rational parameter is bounded under a threshold, then majority voting is the optimal aggregation rule and (2) the group collective decision can be benefited from bounded rationality of individuals.

**Strengths:**

The technical contribution is sound. The results are quite interesting.

**Weaknesses:**

It is not very clear why LLMs are generally considered as quantal best responder

**Questions:**

The theoretical results in this paper generally look great and insightful. However, I wonder the authors could comment more about why LLMs are considered quantal best responder, as section 4.1 only concerns on one domain. Also, is here every LLM considered to have the same \lambda?

---

> ### Author Response · Authors · 2025-11-20
>
> We thank Reviewer shYv for your valuable feedback and positive assessment of our work. We appreciate the opportunity to clarify the connection between LLMs and quantal response.
>
> #### [Q1] Connection between LLMs and Quantal Response
>
> > **Reviewer's Questions:** "I wonder the authors could comment more about why LLMs are considered quantal best responder, as section 4.1 only concerns on one domain. Also, is here every LLM considered to have the same $\lambda$?"
>
> **Our Response:** These are excellent questions. Our reasoning for modeling LLMs as quantal response (QR) agents is based on both a theoretical link and empirical evidence, and we do not assume a single $\lambda$.
>
> **Theoretical Link: QR Model and LLM Softmax**
> The QR model is structurally identical to the LLM's core output mechanism (softmax with temperature). We use $u_j = E[u(X=j, \omega)|S_i]$ to denote the expected utility of choosing action $j$:
>
> * **Quantal Response (QR) Model:** The probability of choosing action $j$ is given by:
>   $$P(X=j) = \frac{e^{\lambda u_j}}{\sum_k e^{\lambda u_k}}$$
>   where $\lambda$ is the rationality parameter.
>
> * **LLM Softmax Mechanism:** The probability of outputting the token (choice) $j$ is determined by the logits $z_j$:
>   $$P(\text{choice } j) = \frac{e^{z_j / t}}{\sum_k e^{z_k / t}}$$
>   where $t$ is the temperature.
>
> These formulations are structurally identical. The rationality parameter $\lambda$ is mathematically analogous to the inverse temperature $(1/t)$. A lower $\lambda$ (more stochasticity) corresponds to a higher temperature $t$ (more randomness). If the LLM’s internal logits are proportional to the expected utility under a given utility function, then the two models coincide.
>
> This connection between stochastic decision models and LLM behavior is an emerging area of research. Recent work has applied QR frameworks to model bounded-rational LLM agents in behavioral game theory. For example, Jia et al. [2025, *NeurIPS*] employ a quantal response framework to quantify and characterize LLMs’ strategic decision-making. Similarly, Kirshner et al. [2025, *Decision Analysis*] use a quantal response framework to model LLM decision-making and conduct a series of experiments with GPT-4, GPT-4o, and o3-mini to test their hypotheses.
>
> **Empirical Validation and $\lambda$ Parameterization (Section 4.1)**
> The logits produced by an LLM are not necessarily proportional to the true expected utility of a decision. To explicitly test the Quantal Response (QR) hypothesis despite this, we designed the experiment in Section 4.1. As shown in Figure 2, the LLM's decision probabilities, plotted against the true expected utility (defined as the probability of matching the ground truth), closely align with the QR model.
>
> **Regarding $\lambda$:** We do not assume a universal $\lambda$. The effective rationality parameter $\lambda$ is a fitted property dependent on **both** the specific model **and** the temperature $t$ used during inference. Our experiment demonstrated this dependency:
>
> * At $t=0.0$, the behavior is deterministic ($\lambda \to \infty$).
> * At $t=0.5$, the model's behavior fits a QR model with $\lambda \approx 13.25$.
> * At $t=1.0$, the model's behavior fits with $\lambda \approx 8.93$.
>
> We also note that the fitted $\lambda$ values exhibit a strong inverse correlation with the temperature $t$, confirming the theoretical analogy where $\lambda \propto 1/t$.
>
> **Regarding "one domain" Generalizability**
> The reviewer is correct that Section 4.1 focuses on a single, controlled Bayesian domain. We chose this domain because it allows us to precisely control the ground truth posterior probability $p$ (which determines expected utility $u_j$), making it possible for the validation of the QR functional form.
>
> Accordingly, Section 4.1 uses the controlled Bayesian environment to verify that the QR model behaves as predicted. Although we cannot perform the same validation on MathQA, since it does not provide a controlled ground truth posterior probability, the aggregation results in Section 4.2 align with the predictions of the QR framework established in Section 4.1. We believe this finding is generalizable because the softmax/temperature mechanism is the fundamental output layer of the LLM used across all domains and its formula is closely related to the formula of the QR model.
>
> We hope this clarifies the link between LLM behavior and the quantal response model, and we will emphasize these points more clearly in the final version. Thanks for your comments!
>
> **References:**
>
> 1. Jia, Jingru, et al. "LLM Strategic Reasoning: Agentic Study through Behavioral Game Theory." *The Thirty-ninth Annual Conference on Neural Information Processing Systems*.
> 2. Kirshner, S. N., Y. Pan, J. X. Wu. 2025. "Prosocial When Simple and Cold-Hearted When Complex: How Task Difficulty Shapes LLM Behavior." *Decision Analysis*. [https://doi.org/10.1287/deca.2025.0396](https://doi.org/10.1287/deca.2025.0396)

---

### Official Review · Reviewer_vc8K · 2025-11-08

**Soundness:** 4
**Presentation:** 4
**Contribution:** 3
**Rating:** 8
**Confidence:** 3

**Summary:**

This work shows that when experts make rational, stochastic decisions (modeled by quantal response), majority voting optimally aggregates their judgments under minimax regret. Surprisingly, such partially random groups can outperform perfectly rational agents, as their randomness preserves informative variation. Experiments with LLMs confirm that aggregating moderately stochastic outputs improves reasoning, revealing bounded rationality as a strength in collective intelligence.

**Strengths:**

1. Theoretical results: the authors proved interesting theoretical results showing that bounded rationality can outperform perfect rationality;
2. Interesting experiments: the authors leverage LLMs as conditional independent agents, which provides a scenario where the condition of the theoretical results can be easily satisfied.

**Weaknesses:**

1. It seems not straightforward to calculate g(n) in the main theorem. Appendix A2 provides a plot of g(n), but mostly observations. May provide more insight on why g(n) = g(n-1) for even n's.
2. (minor) the authors refer to Fig 5 in line 340, but Fig 5 is in the appendix. Better refer to both the appendix section and the figure.

**Questions:**

See above

---

> ### Author Response · Authors · 2025-11-20
>
> We sincerely thank Reviewer vc8K for your positive assessment and for highlighting the strengths of our theoretical results and experiments. We are grateful for the insightful question regarding the observation that $g(n)=g(n-1)$ for even $n$. This is indeed a key property, and we are happy to provide more intuition.
>
> #### [W1] Intuition on $g(n)=g(n-1)$ for even $n$
>
> > **Reviewer's Comment:** "May provide more insight on why $g(n)=g(n-1)$ for even $n$'s."
>
> **Our Response:** Thank you for this excellent question. We will provide intuitions for why $g(n)=g(n-1)$ for even $n$, and $g(n) \neq g(n-1)$ for odd $n$.
>
> **Why $g(2k)=g(2k-1)$?**
> The underlying reason for $g(n)=g(n-1)$ when $n=2k$ (i.e., $n$ is even) is that the expected utility of the majority vote aggregator ($f_{maj}$) is identical for a group of $n=2k$ experts and a group of $n=2k-1$ experts.
>
> In detail, the threshold $g(n)$ is defined as the maximal rationality level for which $f_{maj}$ is optimal. If the performance of $f_{maj}$ is the same at $2k$ and $2k-1$, the threshold $g(n)$ is also the same.
>
> To see why the performance is identical, we can analyze the behavior of the majority vote $f_{maj}$ in both cases:
>
> * **For $n=2k-1$ (odd):** No ties are possible. The majority vote $f_{maj}$ is simply the decision (0 or 1) that has at least $k$ votes.
> * **For $n=2k$ (even):** Ties are possible. The $f_{maj}$ rule is:
>   1. Choose '1' if the number of '1' votes is $x > k$.
>   2. Choose '0' if the number of '1' votes is $x < k$.
>   3. Choose '0' or '1' with 0.5 probability each if $x = k$ (a perfect tie).
>
> The key insight is that the decision process for $n=2k$ experts is equivalent in expectation to randomly removing one expert and taking the majority vote of the remaining $2k-1$ experts. This equivalence holds because:
>
> 1. If the $2k$ experts **are not** in a tie, removing any single expert will not change the majority outcome.
> 2. If the $2k$ experts **are** in a perfect $k-k$ tie, removing one expert (a '0' or a '1' voter) results in a $2k-1$ group whose majority vote will be '1' or '0' with 0.5 probability, respectively.
>
> This perfectly matches the 0.5 random tie-breaking rule defined for $f_{maj}$ when $n=2k$. Since the expected outcome is identical in all situations (tie or no-tie), the total expected utility $U(f_{maj}, \theta)$ is the same for $n=2k$ and $n=2k-1$ for any given report structure $\theta$. This equivalence is also formally demonstrated for a specific signal structure in Lemma A.15.
>
> **Why $g(2k) \neq g(2k+1)$?**
> This symmetry breaks down when comparing $n=2k+1$ (odd) and $n=2k$ (even). Consider a near-tie vote outcome:
>
> * In the $n=2k+1$ case, a vote count of $(k+1)$ '1's and $k$ '0's means the majority aggregator must return '1'.
> * In the $n=2k$ case, when we remove one '1' voter from this $(k+1, k)$ outcome, the remaining $2k$ experts are now in a perfect $k-k$ tie, forcing the $n=2k$ aggregator to randomly return '1' or '0' with 0.5 probability.
>
> Because the $n=2k+1$ aggregator can make a deterministic decision (like choosing '1' in the $(k+1, k)$ case) where the $n=2k$ aggregator would be forced into a stochastic decision (the 0.5 tie-breaking rule), their expected utilities are generally different. This difference in behavior is why $g(2k+1) \neq g(2k)$.
>
> We will also add a note to the appendix (near Figure 4 and Lemma A.15) to clarify this intuition for the final version.
>
> #### [W2] Minor Correction: Figure Reference
>
> Thank you for pointing out the reference issue. We agree that a more precise reference is necessary since Figure 5 is located in the Appendix. We will update the reference in line 340 from "Fig. 5" to a clearer reference in the revised version.
>
> Thank you again for prompting this valuable clarification.

---

### Meta-Review · Area_Chair_ajJA · 2025-12-29

**Summary:**

This paper presents theoretical results on the decision aggregation under uncertainty when stochastic decisions (modeled by quantal response - a stochastic choice model capturing bounded rationality) is considered. The result is that majority voting is the optimal aggregation rule under minimax regret under a certain threshold and group collective quantal decision can outperform perfectly rational agents (ie without stochastic decisions). Experimental evaluation shows that LLMs naturally exhibits quantal response, and moderate aggregation improves accuracy on complex reasoning tasks showing that bounding rationality is able to improve performance.

-vc8K idenfies as strengths the theoretical results and the interestingness of the experiments.
On the other hand, he is skeptical on the computation of the quantity $g(n)$ in the main theorem.

-shYv mentions as positive points the soundness of the technical contribution and the results that are quite interesting.
On the other hand, he is skeptical on the fact that LLMs can be generally considered as quantal best responder.

-eia5 mentions that the paper  offers a valuable contribution to the information aggregation literature, that the results are both elegant and insightful. Experiments are also convincing.
On the other hand, he is not convinced by the second main result (bounded rational advantage) since he is not that convinced by the relevance on the assumption that agents vote sincerely.  He is nevertheless positive on the paper.


This paper has been evaluated positively. All reviewers highlighted this contribution as a valuable one, interesting with good empirical results.
Authors have mainly one weakness to address and they provided rather detailed answers which appear to be relevant.
I propose acceptance based on the elements above.
Nevertheless, I am not an expert of the field and the reviews of vc8K and shYv were rather small which makes me a bit uncertain.

**Reviewer Concerns:**

-vc8K has an answer on his main weakness on the parameter $g(n)$, the answer appears to be well justified.

-shYv received a rather complete answer on his unique main weakness.

-eia5 received an answer on his unique weakness, the answer was rather detailed.

**Reviewer Scores:**

-vc8K gave a 8 so he really supports the paper.

-shYv gave a 6, the answer provided was rather detailed so I may think he would be willing to raise his score

-eia5 gave a 6. It is unclear is the answer would be fully convincing for the reviewer, but it is at least well argumented.

---

### Decision · Program_Chairs · 2026-01-26

Accept (Poster)